# Rethinking the Trust Region in LLM Reinforcement Learning

Penghui Qi [* 1 2]  Xiangxin Zhou [* 1]  Zichen Liu [2]  Tianyu Pang [1]  Chao Du [1]  Min Lin [1]  Wee Sun Lee [2]

## Abstract

Reinforcement learning (RL) has become a cornerstone for fine-tuning Large Language Models (LLMs), with Proximal Policy Optimization (PPO) serving as the de facto standard algorithm. Despite its ubiquity, we argue that the core ratio clipping mechanism in PPO is structurally ill-suited for the large vocabularies inherent to LLMs. PPO constrains policy updates based on the probability ratio of sampled tokens, which serves as a noisy single-sample Monte Carlo estimate of the true policy divergence. This creates a sub-optimal learning dynamic: updates to low-probability tokens are aggressively over-penalized, while potentially catastrophic shifts in high-probability tokens are under-constrained, leading to training inefficiency and instability. To address this, we propose **Divergence Proximal Policy Optimization (DPPO)**, which substitutes heuristic clipping with a more principled constraint based on a direct estimate of policy divergence (e.g., Total Variation or KL). To avoid huge memory footprint, we introduce the efficient Binary and Top-K approximations to capture the essential divergence with negligible overhead. Extensive empirical evaluations demonstrate that DPPO achieves superior training **stability** and **efficiency** compared to existing methods, offering a more robust foundation for RL-based LLM fine-tuning. Our code is available at https://github.com/sail-sg/Stable-RL.

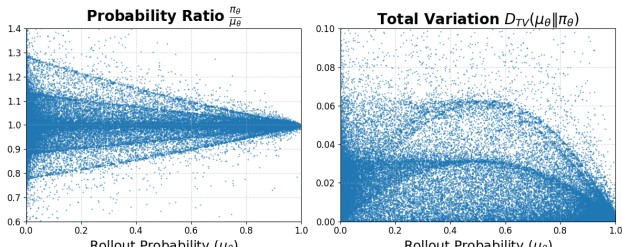

*Figure 1.* The plots show numerical differences between training and inference engines for Qwen3-30B-A3B-Base with identical parameters. (**Left**) The token-level probability ratio (used in PPO) is highly volatile for low-probability tokens. (**Right**) The token-level TV divergence (used in DPPO) is more stable. This highlights a key flaw of PPO's clipping: it over-penalizes low-probability tokens, which can slow down learning; and under-penalizes high-probability tokens, which can permit large, destabilizing updates.

## 1. Introduction

Reinforcement learning (RL) is a foundational paradigm for fine-tuning Large Language Models (LLMs), enabling alignment with human preferences (Ouyang et al., 2022; Rafailov et al., 2023) and complex reasoning tasks (Guo et al., 2025; Qi et al., 2025a). In practice, LLM RL is often off-policy: an inference engine samples rollouts, while a trainer engine computes gradients. Even when the two engines use the same model parameters, *training-inference mismatch* (Yao et al., 2025; Qi et al., 2025b; Zheng et al., 2025) can make their token distributions differ. Practical systems also use collected rollouts for several minibatch updates to improve throughput (Liu et al., 2025a). These choices make it essential to control how far each update moves the trained policy from the policy that generated the data.

Proximal Policy Optimization (PPO[1]) (Schulman et al., 2017) is the dominant algorithm in this setting because it is simple and scalable. Its main safeguard is ratio clipping. If the probability ratio between the new policy and the behavior policy becomes too large or too small on a sampled token, PPO clips the objective. This rule is meant to keep updates inside a *trust region*, where monotonic improvement is theoretically guaranteed (Schulman et al., 2015).

Despite its widespread adoption, we argue that PPO's core mechanism, ratio clipping, is structurally ill-suited for the expansive, long-tailed vocabularies inherent to LLMs. Although motivated by Trust Region Policy Optimization (TRPO) (Schulman et al., 2015), which constrains KL or Total Variation (TV) divergence of policy distributions, PPO

---

[*]Equal contribution  [1]Sea AI Lab, Singapore  [2]School of Computing, National University of Singapore. Correspondence to: Wee Sun Lee <leews@comp.nus.edu.sg>, Penghui Qi <penghuiq@comp.nus.edu.sg>.

*Proceedings of the 43rd International Conference on Machine Learning*, Seoul, South Korea. PMLR 306, 2026. Copyright 2026 by the author(s).

---

[1]We denote PPO by its ratio-clipping loss, regardless of advantage estimation. Under this definition, GRPO is a PPO variant.

instead clips the probability ratio of the sampled token. This ratio is only a noisy, single-sample estimate of the true policy divergence. While this approximation may suffice for classical RL environments with limited action spaces, it fails in the LLM regime because the ratio depends strongly on the original token probability. For example, increasing a rare token's probability from $10^{-5}$ to $10^{-3}$ gives a ratio of 100 and triggers clipping, although the moved probability mass is small. Conversely, decreasing a high-probability token from 0.99 to 0.8 moves much more mass, yet the ratio may remain inside a typical clipping range.

Training-inference mismatch makes this problem worse. As illustrated in Figure 1, the probability ratio becomes highly volatile for low-probability tokens, while TV divergence remains stable. Consequently, PPO creates a sub-optimal learning dynamic: updates to low-probability tokens are aggressively over-penalized, slowing learning, while updates to high-probability tokens are under-penalized, risking instability. This implicit bias necessitates a fundamental rethinking of the trust region approach in LLM fine-tuning to ensure both efficiency and stability.

To address this limitation, we propose Divergence Proximal Policy Optimization (DPPO), which replaces ratio-based clipping with a constraint based on policy divergence. Rather than relying on noisy single-sample ratios, DPPO directly estimates policy divergence (e.g., TV or KL divergence). To ensure memory feasibility, we introduce two efficient approximations, Binary and Top-K divergence, which capture essential distributional shifts with negligible overhead. This allows DPPO to rigorously distinguish between safe and unsafe updates, effectively resolving the problems of over- and under-constraining inherent in standard PPO.

In this work, we provide a comprehensive rethinking of the trust region in the context of LLM fine-tuning. Our contributions are threefold. **Theoretical Formulation**: We derive policy improvement bounds specifically tailored to the finite-horizon, undiscounted setting of LLM generation, establishing a rigorous theoretical foundation for trust-region methods in this domain. **Stability and Efficiency Analysis**: We isolate the primary sources of training instability to provide practical stabilization guidelines, while further highlighting the significant role that low-probability tokens play in driving exploration. **Algorithmic Performance**: We demonstrate that DPPO achieves superior stability and final performance compared to existing methods like GRPO, providing a robust new framework for RL-based fine-tuning.

## 2. Background

### 2.1. Policy Performance Difference

We begin with the standard formulation of a Markov Decision Process (MDP), defined by the tuple $\mathcal{M} =$

$(\mathcal{S}, \mathcal{A}, P, r, \rho_0, \gamma)$, which includes the state space $\mathcal{S}$, action space $\mathcal{A}$, transition dynamics $P(s'|s, a)$, reward function $r(s, a)$, initial state distribution $\rho_0(s)$, and a discount factor $\gamma \in [0, 1]$. A stochastic policy $\pi(a|s)$ generates trajectories $\tau = (s_0, a_0, r_0, s_1, a_1, r_1, \ldots)$ by sampling actions $a_t \sim \pi(\cdot|s_t)$ and transitioning to states $s_{t+1} \sim P(\cdot|s_t, a_t)$. The central goal of RL is to find a policy that maximizes the expected discounted return:

$$\eta(\pi) = \mathbb{E}_{\tau \sim \pi} \left[ \sum_{t=0}^{\infty} \gamma^t r_t \right].$$

To facilitate policy optimization, we define the standard value functions under a policy $\pi$: the state-value function $V^\pi(s) = \mathbb{E}_{\tau \sim \pi} \left[ \sum_{t=0}^{\infty} \gamma^t r_t \big| s_0 = s \right]$, the action-value function $Q^\pi(s, a) = \mathbb{E}_{\tau \sim \pi} \left[ \sum_{t=0}^{\infty} \gamma^t r_t \big| s_0 = s, a_0 = a \right]$, and the advantage function $A^\pi(s, a) = Q^\pi(s, a) - V^\pi(s)$. A key theoretical tool for relating the performance of two distinct policies is the policy performance difference theorem (Kakade & Langford, 2002). It states that for any two policies, a target policy (to be optimized) $\pi$ and a behavior policy (for rollout) $\mu$, their expected returns are related by:

$$\eta(\pi) - \eta(\mu) = \frac{1}{1 - \gamma} \mathbb{E}_{s \sim \rho^\pi, a \sim \pi(\cdot|s)} \left[ A^\mu(s, a) \right]. \quad (1)$$

Here, $\rho^\pi(s) = (1 - \gamma) \sum_{t=0}^{\infty} \gamma^t \Pr(s_t = s | \pi)$ is the normalized discounted state-visitation distribution induced by the policy $\pi$. This identity is fundamental, as it implies that any policy update that results in a non-negative expected advantage guarantees monotonic performance improvement, i.e., $\eta(\pi) \geq \eta(\mu)$.

### 2.2. Policy Improvement Bound

While Equation 1 provides a direct expression for policy improvement, its dependence on the state-visitation distribution $\rho^\pi$ of the new policy makes it intractable for direct optimization. To overcome this, Schulman et al. (2015) derive a lower bound on performance improvement that can be estimated using samples from the behavior policy $\mu$, with a penalty term that measures the divergence between the old and new policies. This lower bound forms the basis of trust-region methods.

**Theorem 2.1.** *(Schulman et al., 2015; Achiam et al., 2017) Given any two policies, $\mu$ and $\pi$, the following bound holds:*

$$\eta(\pi) - \eta(\mu) \geq \frac{1}{1 - \gamma} \mathbb{E}_{s \sim \rho^\mu, a \sim \mu(\cdot|s)} \left[ \frac{\pi(a|s)}{\mu(a|s)} A^\mu(s, a) \right]$$

$$- \frac{2\xi\gamma}{(1 - \gamma)^2} D_{\mathrm{TV}}^{\max}(\mu \| \pi)^2, \quad (2)$$

*where $\xi = \max_{s,a} \left| A^\mu(s, a) \right|$ and $D_{\mathrm{TV}}^{\max}(\mu \| \pi) = \max_s D_{\mathrm{TV}}\left( \mu(\cdot|s) \| \pi(\cdot|s) \right)$, which is the maximum Total Variation (TV) divergence among all states.*

This bound provides a direct path to guaranteed policy improvement. The right-hand side of the inequality forms a surrogate objective that is a tight lower bound on the true performance improvement, touching the objective when $\pi = \mu$. Therefore, iteratively maximizing this surrogate guarantees monotonic improvement in $\eta(\pi)$, following the principles of the Minorize-Maximization (MM) algorithm (Hunter & Lange, 2004; Schulman et al., 2015).

### 2.3. Trust Region Policy Optimization

The policy improvement bound in Equation (2) directly justifies a surrogate objective,

$$ L_\mu(\pi) = \frac{1}{1-\gamma} \mathbb{E}_{s\sim\rho^\mu, a\sim\mu(a|s)} \left[ \frac{\pi(a|s)}{\mu(a|s)} A^\mu(s,a) \right]. \quad (3) $$

This objective serves as a **first-order approximation** of the true performance improvement $\eta(\pi) - \eta(\mu)$, as their values and gradients match at the point of expansion $\pi = \mu$ (Kakade & Langford, 2002; Schulman et al., 2015; Zheng et al., 2025). Therefore, maximizing $L_\mu(\pi)$ within a small *trust region* guarantees stable and meaningful policy improvement. This insight motivates the trust-region optimization approach (Schulman et al., 2015; Xie et al., 2024), which involves maximizing $L_\mu(\pi)$ subject to a constraint that keeps the new policy $\pi$ within a trust region around the current policy $\mu$, thereby ensuring the validity of the approximation. Formally, this is expressed as the following constrained optimization problem:

$$ \max_\pi \quad L_\mu(\pi), \qquad \text{s.t.} \quad D_{\text{TV}}^{\max}(\mu \,\|\, \pi) \leq \delta, \quad (4) $$

where the constraint can also be applied on a KL divergence $D_{\text{KL}}$, justified via Pinsker's inequality:

$$ D_{\text{TV}}(\mu \,\|\, \pi)^2 \leq \tfrac{1}{2} D_{\text{KL}}(\mu \,\|\, \pi). $$

## 3. Trust Region Under LLM Regime

In this section, we adapt the trust region framework to the specific context of LLM fine-tuning. This setting differs from the classical RL paradigm in two crucial ways. First, the learning problem is structured as an undiscounted ($\gamma = 1$) episodic task with a finite horizon $T$, which makes the original bound in Equation (2) ill-defined, as the $\frac{1}{1-\gamma}$ term diverges to infinity. Second, due to the sparse reward nature, advantages are often estimated at the sequence level (Shao et al., 2024), rather than on a per-token basis.

Formally, given a prompt $x$, a policy $\pi$ (the LLM) generates a response $y = (y_1, \ldots, y_T)$ by sequentially sampling tokens. At each step $t$, the policy defines a conditional distribution $\pi(y_t|s_t)$ over the vocabulary $\mathcal{A}$, where the state $s_t = (x, y_1, \ldots, y_{t-1})$ consists of the prompt and previously generated tokens. The probability of the complete response is the product of these conditional probabilities: $\pi(y|x) = \prod_{t=1}^{T} \pi(y_t|s_t)$. After the full response is generated, a scalar reward $R(y,x)$ is provided. For brevity, we will omit the dependency on the initial prompt $x$ and write the objective function as:

$$ \mathcal{J}(\pi) = \mathbb{E}_{y\sim\pi}[R(y)]. $$

We now derive performance difference identity and policy improvement bound tailored to this regime.

**Theorem 3.1** (Performance Difference Identity for LLMs). *In a finite-horizon setting ($T$) with no discount ($\gamma = 1$), for any two policies $\pi$ and $\mu$, the performance difference can be decomposed as:*

$$ \mathcal{J}(\pi) - \mathcal{J}(\mu) = L'_\mu(\pi) - \Delta(\mu, \pi), $$

*where $L'_\mu(\pi)$ is a surrogate objective defined as:*

$$ L'_\mu(\pi) = \mathbb{E}_{y\sim\mu} \left[ R(y) \sum_{t=1}^{|y|} \left( \frac{\pi(y_t|s_t)}{\mu(y_t|s_t)} - 1 \right) \right], \quad (5) $$

*and $\Delta(\mu, \pi)$ is an error term given by:*

$$ \Delta(\mu, \pi) = \mathbb{E}_{y\sim\mu} \Big[ R(y) \qquad\qquad (6) $$
$$ \sum_{t=1}^{|y|} \left( \frac{\pi(y_t|s_t)}{\mu(y_t|s_t)} - 1 \right) \left( 1 - \prod_{j=t+1}^{T} \frac{\pi(y_j|s_j)}{\mu(y_j|s_j)} \right) \Big]. $$

This theorem provides an exact expression for the policy improvement. The surrogate $L'_\mu(\pi)$ represents a first-order approximation, while the error term $\Delta$ captures the higher-order effects of the policy change. To make this practical for optimization, we bound the error term.

**Theorem 3.2** (Policy Improvement Bound for LLMs). *In a finite-horizon setting ($T$) with no discount ($\gamma = 1$), let $\xi = \max_y |R(y)|$ be the maximum absolute reward, $D_{\text{TV}}^{\max}(\mu \,\|\, \pi) = \max_{s_t} D_{\text{TV}}\big(\mu(\cdot|s_t) \,\|\, \pi(\cdot|s_t)\big)$ be the maximum Total Variation (TV) divergence over all states, and $\bar{D}_{\text{TV}}(\mu, \pi) = \mathbb{E}_{y\sim\mu}\big[\sum_{t=1}^{|y|} D_{\text{TV}}\big(\mu(\cdot|s_t) \,\|\, \pi(\cdot|s_t)\big)\big]$ be the average token-level TV divergence. Then the policy improvement is lower-bounded by both:*

$$ \mathcal{J}(\pi) - \mathcal{J}(\mu) \geq L'_\mu(\pi) - 2\xi T(T-1) \cdot D_{\text{TV}}^{\max}(\mu \,\|\, \pi)^2, \quad (7) $$
$$ \mathcal{J}(\pi) - \mathcal{J}(\mu) \geq L'_\mu(\pi) - 4\xi \bar{D}_{\text{TV}}(\mu, \pi). \quad (8) $$

This theorem establishes lower bounds on policy improvement. The max-divergence form in Equation (7) is structurally analogous to the bound in Theorem 2.1 (see Appendix B.4), with the horizon $T$ playing a role similar to the effective horizon $\frac{1}{1-\gamma}$ in the discounted setting. The average-divergence form in Equation (8) is tighter for long

LLM responses and more directly matches the per-token divergence control used in PPO and our algorithm. Together, they provide a clear theoretical justification for adapting the trust region approach into LLM regime. Similar to Equation (4), we can solve the following constrained optimization problem to guarantee stable learning:

$$\max_{\pi} \quad L'_\mu(\pi), \qquad \text{s.t.} \quad D_{\text{TV}}^{\max}(\mu \,\|\, \pi) \leq \delta, \qquad (9)$$

where the constraint can also be applied on a KL divergence.

The proofs for Theorem 3.1 and both parts of Theorem 3.2 are deferred to Appendix B.

# 4. Methodology

## 4.1. Proximal Policy Optimization

While theoretically appealing, the constrained optimization in TRPO requires second-order information and is difficult to scale. PPO (Schulman et al., 2017) was introduced as a first-order alternative that retains much of the stability. Owing to its simplicity and strong empirical performance, PPO has become a standard algorithm for fine-tuning LLMs.

Instead of enforcing an explicit trust-region constraint, PPO optimizes a clipped surrogate objective:

$$L_\mu^{\text{PPO}}(\pi) = \mathbb{E}_{y\sim\mu}\left[\sum_{t=1}^{|y|}\min\left(r_t\hat{A}_t, \text{clip}(r_t, 1-\epsilon, 1+\epsilon)\hat{A}_t\right)\right],$$
$$r_t = \frac{\pi(y_t|s_t)}{\mu(y_t|s_t)}, \qquad (10)$$

where $\hat{A}_t$ is an estimated advantage at timestep $t$. In LLM fine-tuning, it is often estimated as $\hat{A} = R(y) - \frac{1}{G}\sum_{i=1}^{G} R(y_i)$ (Shao et al., 2024; Liu et al., 2025d), where $\{y_i\}_{i=1}^{G}$ is a group of responses for the same prompt.

The term $r_t\hat{A}_t$ in Equation (10) is the commonly used form of the surrogate in Equation (5): replacing $r_t - 1$ with $r_t$ leaves the gradient unchanged, while replacing $R(y)$ with $\hat{A}_t$ reduces variance without biasing the expected policy gradient (see Appendix B.4). The combination of the $\min$ and $\text{clip}$ operations then acts as an implicit trust-region constraint. Once $r_t$ is outside the interval $[1 - \epsilon, 1 + \epsilon]$, the clipped branch removes the incentive to move the new policy farther from the old one. The connection between this clipping rule and the formal trust region can be understood by examining the TV divergence:

$$D_{\text{TV}}\big(\mu(\cdot|s_t)\,\|\,\pi(\cdot|s_t)\big) = \frac{1}{2}\mathbb{E}_{y_t\sim\mu}\left[\,|r_t - 1|\,\right]. \qquad (11)$$

From this perspective, PPO's clipping condition, $|r_t - 1| \leq \epsilon$, can be interpreted as constraining a **single-sample Monte Carlo estimate** of the expected value in Equation (11). In essence, PPO enforces its trust region not on the true TV divergence, but on a noisy, single-point estimation. As we will argue next, this crude approximation is the source of significant pathologies when applied to the large, long-tailed vocabulary distributions characteristic of LLMs.

## 4.2. Limitations of PPO Ratio Clipping

The key limitation of PPO is that whether an update is clipped depends heavily on the sampled token's probability, rather than the true TV divergence between $\mu(\cdot|s_t)$ and $\pi(\cdot|s_t)$. Concretely, consider a fixed state $s$ and two tokens $a_{\text{low}}$ and $a_{\text{high}}$ with

$$\mu(a_{\text{low}}|s) = 10^{-4}, \qquad \pi(a_{\text{low}}|s) = 10^{-2},$$
$$\mu(a_{\text{high}}|s) = 0.99, \qquad \pi(a_{\text{high}}|s) = 0.80.$$

The probability ratio for the low-probability token is $r_{\text{low}} = \frac{10^{-2}}{10^{-4}} = 100$, which is far outside a typical clipping range $[1-\epsilon, 1+\epsilon]$ (e.g., $\epsilon = 0.2$). PPO would thus heavily clip the contribution of this update. In contrast, the actual contribution of this change to the TV divergence can be very small, because the total mass moved at $a_{\text{low}}$ is tiny. For the high-probability token, $r_{\text{high}} = \frac{0.80}{0.99} \approx 0.808$, which can still lie *inside* the clipping range for a moderate $\epsilon$. Yet this update removes 0.19 probability mass from the dominant token, and therefore induces a much larger contribution to $D_{\text{TV}}$.

These examples highlight a structural flaw in PPO's clipping heuristic. For **low-probability tokens**, an update that produces a large probability ratio is aggressively constrained, even when its impact on the TV divergence is negligible, thereby slowing training efficiency. Conversely, for **high-probability tokens**, an update producing a ratio close to one may go unpenalized, even when the absolute change in probability mass is large enough to cause a substantial TV divergence, which in turn risks training instability.

**Connections to Existing Work** The insight that PPO's ratio clipping disproportionately penalizes low-probability tokens aligns with several prior studies. For instance, methods like *Clip-Higher* (Yu et al., 2025) and CISPO (Chen et al., 2025) observe that important "exploration" or "reasoning" tokens often have low initial probabilities (see Appendix E). These tokens usually get high importance ratios during policy updates and are consequently clipped, hindering the learning process. However, the solutions proposed remain heuristic and problematic. *Clip-Higher* suggests manually increasing the upper clipping bound, while CISPO continues to apply the gradient even for large divergence, completely ignoring the trust region. While these methods correctly identify the symptom, they fail to address the root cause: the fundamental mismatch between the single-sample probability ratio and the true distributional divergence.

### 4.3. Divergence Proximal Policy Optimization

To address the limitations of PPO's ratio clipping, we introduce Divergence Proximal Policy Optimization (DPPO), which directly uses a divergence-based constraint grounded in trust region theory. Similar to PPO, we employ a dynamic mask to block updates that would push policy outside the trust region. The DPPO objective is:

$$L_\mu^{\text{DPPO}}(\pi) = \mathbb{E}_{y \sim \mu} \left[ \sum_{t=1}^{|y|} M_t^{\text{DPPO}} \cdot r_t \cdot \hat{A}_t \right],$$

$$M_t^{\text{DPPO}} = \begin{cases} 0, & \text{if } (\hat{A}_t > 0 \text{ and } r_t > 1 \text{ and } D > \delta) \text{ or} \\ & \quad (\hat{A}_t < 0 \text{ and } r_t < 1 \text{ and } D > \delta) \\ 1, & \text{otherwise}, \end{cases} \quad (12)$$

where $D \equiv D\big(\mu(\cdot|s_t) \| \pi(\cdot|s_t)\big)$ denotes the divergence (e.g., TV or KL) between the rollout and training policy distributions, and $\delta$ is a divergence threshold hyperparameter. As noted by Chen et al. (2025); Zheng et al. (2025), this objective recovers the original PPO algorithm in Equation (10) when the divergence $D$ is replaced by $|r_t - 1|$. Our key innovation lies in the design of this mask: instead of relying on the noisy single-sample ratio, it is conditioned on a direct measure of the distributional policy shift.

This design directly approximates the formal trust region constraint from Theorem 3.2 while preserving the beneficial asymmetric structure of PPO's clipping. The mask only considers blocking an update if it is already moving away from the trusted region (i.e., $r_t > 1$ for a positive advantage or $r_t < 1$ for a negative advantage). It never blocks updates that move the policy ratio towards one (e.g., when $\hat{A}_t > 0$ and $r_t < 1$), a desirable property for accelerating learning.

Unlike PPO, the final decision to block an update is based on whether the entire policy distribution has shifted too far ($D > \delta$), not on the noisy and often misleading ratio of a single sample. This resolves the over- and under-constraining issues inherent in standard PPO. The primary remaining challenge is the overhead of calculating the full divergence $D$ over a large vocabulary in LLMs, which we address next.

### 4.4. Approximating Distribution Divergence

Directly computing the policy divergence is memory-prohibitive for LLMs. To make it practical, we introduce two lightweight approximations, which serve as principled lower bounds of the true divergence (see Appendix C).

**Binary Approximation**  The binary approximation collapses the original categorical distribution into a simple Bernoulli distribution, distinguishing only between the sampled token and all other tokens. We define the new distribution as: $p_t^{\tilde{\pi}} = \big(\tilde{\pi}(a_t|s_t), \quad 1 - \tilde{\pi}(a_t|s_t)\big)$, where $\tilde{\pi}$ can be $\mu$

or $\pi$. The TV and KL divergences are then computed as:

$$D_{\text{TV}}^{\text{Bin}}(t) = \big| \mu(a_t|s_t) - \pi(a_t|s_t) \big|, \quad (13)$$

$$D_{\text{KL}}^{\text{Bin}}(t) = \mu(a_t|s_t) \log \frac{\mu(a_t|s_t)}{\pi(a_t|s_t)} \\ + (1 - \mu(a_t|s_t)) \log \frac{1 - \mu(a_t|s_t)}{1 - \pi(a_t|s_t)}. \quad (14)$$

This binary divergence can be computed at negligible overhead. Crucially, it correctly distinguishes between large versus small shifts in absolute probability mass, thereby resolving the primary failure mode of PPO's clipping.

**Top-K Approximation**  To provide a richer and more faithful approximation of the distributional shift, the top-K variant explicitly tracks the most probable tokens. First, we define a small, representative set of tokens $\mathcal{A}_t'$ as: $\mathcal{A}_t' = \text{TopK}\big(\mu(\cdot|s_t), K\big) \cup \{a_t\}$, which includes the $K$ highest-probability tokens under the behavior policy, augmented with the sampled token $a_t$ if it is not already present. We then form reduced categorical distributions, $p_t^\mu$ and $p_t^\pi$, over the new vocabulary $\mathcal{A}_t'' = \mathcal{A}_t' \cup \{\text{other}\}$. For any token $a \in \mathcal{A}_t'$, its probability is its original probability, while all other tokens are aggregated into the "other" category: $p_t^{\tilde{\pi}}(a) = \tilde{\pi}(a|s_t) \quad \forall a \in \mathcal{A}_t'$, and $p_t^{\tilde{\pi}}(\text{other}) = 1 - \sum_{a \in \mathcal{A}_t'} \tilde{\pi}(a|s_t)$, where $\tilde{\pi}$ can be $\mu$ or $\pi$. The divergence is then computed over this reduced distribution:

$$D_{\text{TV}}^{\text{TopK}}(t) = \frac{1}{2} \sum_{a \in \mathcal{A}_t''} \big| p_t^\mu(a) - p_t^\pi(a) \big|, \quad (15)$$

$$D_{\text{KL}}^{\text{TopK}}(t) = \sum_{a \in \mathcal{A}_t''} p_t^\mu(a) \log \frac{p_t^\mu(a)}{p_t^\pi(a)}. \quad (16)$$

This approach better captures changes in the head of the policy distribution, which typically dominates the true divergence value. The overhead is minimal, making it a practical and high-fidelity choice for DPPO.

## 5. Analysis on Training Stability

In this section, we conduct an empirical study to dissect the training instability issue caused by training-inference mismatch (see Appendix A.2). To formalize our analysis, we denote the parameters being optimized as $\theta$ and the parameters used for data generation as $\theta'$. We aim to answer three fundamental research questions:

1. Given the extremely low learning rates (e.g., $10^{-6}$) common in LLM fine-tuning, is a trust region still necessary to ensure training stability?
2. Should the trust region be defined with respect to the original rollout distribution ($\mu_{\theta'}$) or a recomputed policy distribution ($\pi_{\theta'}$)?
3. What specific types of policy updates are the primary drivers of training instability?

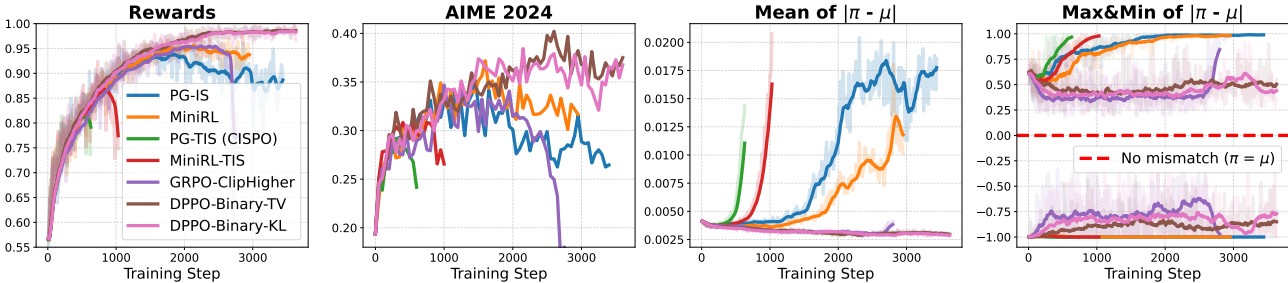

*Figure 2.* DPPO variants achieve stable training while controlling the training-inference mismatch at a low level. In contrast, methods without a trust region (PG-IS, CISPO) or with a misspecified one (MiniRL) suffer from growing mismatch and eventual collapse.

**Experimental Setting:** Our experimental setup follows the sanity test proposed by Qi et al. (2025b). We fine-tune DeepSeek-R1-Distill-Qwen-1.5B (Guo et al., 2025) on a curated set of 1,460 problems from the MATH dataset (Hendrycks et al., 2021). In this setting, a stable algorithm should theoretically converge to 100% training accuracy, as all problems are known to be solvable by the initial model.

We evaluate several algorithms, each representing a different approach to managing the policy update. The baselines include: **PG-IS** and its truncated variant **PG-TIS** (also known as CISPO (Chen et al., 2025)), which use standard policy gradients with token-level importance sampling; **GRPO with Clip-Higher**, a PPO-like algorithm where clipping is based on the rollout policy ratio $r_t = \frac{\pi_\theta}{\mu_{\theta'}}$ (Shao et al., 2024; Liu et al., 2025d); and **MiniRL & MiniRL-TIS**, a PPO variant where clipping is based on a recomputed policy ratio $r_t = \frac{\pi_\theta}{\pi_{\theta'}}$ (Zheng et al., 2025). We compare these against **DPPO**, our proposed method using either binary KL or TV divergence, with the trust region defined with respect to the rollout distribution $\mu_{\theta'}$. Detailed configurations for each algorithm are provided in the Appendix D.

### 5.1. The Necessity of a Trust Region

Our first question addresses whether a trust region is redundant at low learning rates. Figure 2 provides a clear answer. The unconstrained methods, PG-IS and PG-TIS (CISPO), both suffer from an increasing training-inference mismatch, which culminates in a collapse of performance. In contrast, our DPPO variants, which enforce a principled trust region, maintain a stable, low level of mismatch throughout training and achieve near-perfect final rewards.

**Takeaway 1:** A trust region is essential for stable training, even with very small learning rates. Without it, the training-inference mismatch accumulates and leads to collapse.

### 5.2. The Correct Anchor for the Trust Region

Next, we investigate to which distribution the trust region should be anchored. A common practice in open-source implementations (Sheng et al., 2024; Zhu et al., 2025) is to use

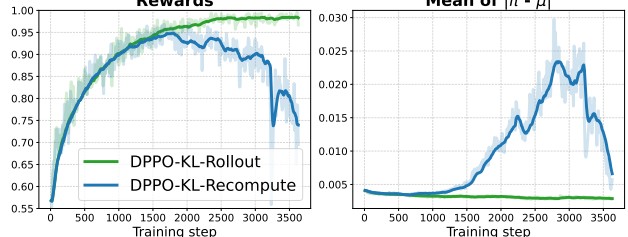

*Figure 3.* Switching the stable DPPO-KL to a decoupled objective causes the mismatch to grow and performance to collapse, confirming that the trust region must be anchored to the rollout policy.

a *decoupled* objective (Hilton et al., 2022), where the trust region is enforced relative to a recomputed policy distribution ($\pi_{\theta'}$) instead of the original behavior policy ($\mu_{\theta'}$). The MiniRL algorithm, for example, follows this design (Zheng et al., 2025). Our results show this choice is detrimental. As in Figure 2, MiniRL fails to control the training-inference mismatch and its performance collapses, despite using a trust region. To confirm this, we created a decoupled version of our stable DPPO-KL algorithm. Figure 3 shows that this single change corrupts the stable training process, causing the mismatch to grow and performance to collapse.

**Takeaway 2:** The trust region must be defined with respect to the original behavior policy ($\mu_{\theta'}$). Using a recomputed on-policy distribution as the anchor leads to instability. This finding aligns with the theoretical bound in Equation (7) and offers a significant practical benefit: by removing the need for recomputation, we can reduce training costs by approximately 25% (Qi et al., 2024).

### 5.3. Identifying the Source of Instability

Finally, we seek to pinpoint which specific policy updates are most responsible for the instability. Our methodology is to start with the unstable PG-IS algorithm, which applies no update masking, and introduce the most minimal mask necessary to restore stability. This allows us to isolate the most detrimental class of updates. Since updates on positively rewarded samples are typically safe, we focus on negative samples where the policy is penalized (Liu et al., 2025a; Ren & Sutherland, 2025). We design a simple mask that only

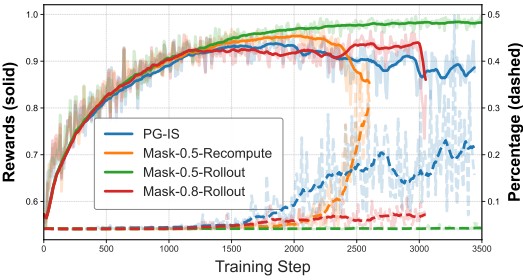

*Figure 4.* Isolating the source of instability. The solid curves are training rewards, while the dashed lines are the percentage of *bad updates*. Starting with the unstable PG-IS, applying a minimal mask that only blocks large-divergence bad updates on negative samples is sufficient to stabilize training, indicating these bad updates are the primary cause of training instability.

blocks updates on negative samples where the probability of the sampled token is decreased by more than a threshold $\delta$: $M_t = 0$ if $\hat{A}_t < 0$ and $\mu_{\theta'}(y_t|s_t) - \pi_\theta(y_t|s_t) \geq \delta$. As shown in Figure 4, applying this minimal mask with $\delta = 0.5$ is sufficient to stabilize the training. In contrast, a slightly looser mask ($\delta = 0.8$) or one anchored to the recomputed distribution ("Mask-0.5-Recompute") both fail to prevent the eventual collapse. We define *bad updates* as those where this divergence exceeds 0.5 and plot their percentage over time. The plot reveals that only a very small fraction of updates are "bad" ($\leq 0.5\%$) yet they are the primary culprits behind training collapse. Furthermore, the percentage of these bad updates strongly correlates with reward fluctuation; as the fraction of bad updates rises, the reward curve becomes more erratic, reinforcing a causal link.

**Takeaway 3:** The primary source of instability is a small subset of updates on negative samples that push the policy far outside the trust region. A likely reason is that aggressively penalizing a token the model deems probable can corrupt the LLM's internal knowledge and destabilize the learning process. This finding confirms the critical need for a trust region, particularly when handling negative feedback.

### 5.4. The Pitfalls of Truncated Importance Sampling

Our results also reveal an unexpected drawback of Truncated Importance Sampling (TIS), a common technique for reducing policy-gradient variance (Yao et al., 2025; Chen et al., 2025). In our experiments, TIS worsens training stability: as shown in Figure 2, PG-TIS and MiniRL-TIS collapse prematurely and substantially underperform their untruncated counterparts. We hypothesize that this detrimental effect stems from the same issue as PPO's ratio clipping: low-probability tokens, which naturally produce high-variance ratios, are the most likely to be truncated by TIS. While this does reduce variance, it systematically down-weights the gradient signal from these tokens, introducing a significant and harmful bias into the policy update. This suggests that naive truncation can be as damaging as naive clipping.

## 6. Analysis on Training Efficiency

Beyond training stability, the design of trust region is also critical for training *efficiency*. As motivated in Section 4.2, PPO's ratio-clipping over-constrains the updates to low-probability tokens, which might be permitted by a divergence-based trust region. In this section, we aim to analyze how low-probability tokens affect the training dynamics, thus justifying the adoption of divergence-based trust region in our DPPO algorithm.

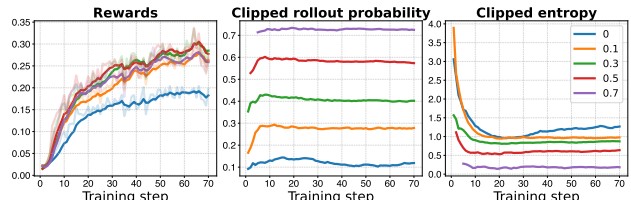

*Figure 5.* Analysis of relaxing trust regions for low-probability tokens. (**Left**) Training reward curves. (**Middle**) Rollout probability of clipped tokens. (**Right**) Entropy of clipped tokens.

**Experimental Setting:** We fine-tune Qwen3-1.7B-Base (Yang et al., 2025) on the DAPO dataset (Yu et al., 2025). We employ GRPO (Guo et al., 2025; Liu et al., 2025d) with the Clip-Higher trick (Yu et al., 2025) as the baseline algorithm. We then *relax* trust regions by setting the clipping threshold $\epsilon$ in Equation (10) as infinity for tokens with $\mu(y_t|s_t) < \alpha$, thus isolating the effect of low-probability tokens.

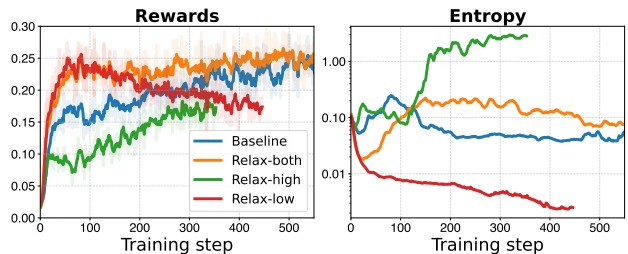

*Figure 6.* Analysis of trust region relaxation direction. (**Left**) Training reward curves. (**Right**) Policy entropy.

The learning curves for varying values of $\alpha$ are presented in Figure 5. Notably, relaxing the clipping constraint for tokens with $\mu(y_t|s_t) < 0.1$ yields a substantial improvement in training efficiency compared to the GRPO baseline ($\alpha = 0$). This observation validates our hypothesis that the ratio-clipping mechanism in PPO over-constrains updates to low-probability tokens, thereby hindering overall learning progress. The middle plot reveals that **clipped tokens are predominantly characterized by low probabilities** (typically below 0.15 for the baseline in blue). As $\alpha$ increases, the probabilities of clipped tokens also rise, confirming that PPO's ratio-clipping is structurally biased against low-probability tokens. Furthermore, the right

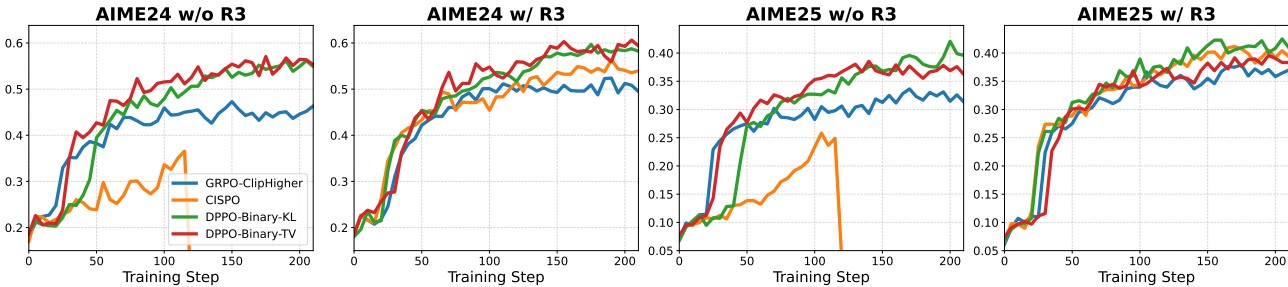

*Figure 7.* Evolution of AIME24 and AIME25 Avg@32 scores during RL training using Qwen3-30B-A3B-Base. The first and third panels correspond to the same experiment without rollout router replay (w/o R3), while the second and fourth panels correspond to the same experiment with rollout router replay (w/ R3).

plot demonstrates that **clipped tokens frequently exhibit high entropy**. Consistent with Wang et al. (2025a), which posits that RL is driven primarily by high-entropy tokens in LLMs, our results suggest that relaxing constraints on these tokens enables more informative policy updates and thus achieves higher training efficiency (see Appendix E for most frequent clipped tokens).

Furthermore, we examine the effect of directional clip relaxation with a fixed $\alpha = 0.1$. We generalize the clip operation with asymmetric thresholds, denoted as $\text{clip}(r_t, 1 - \epsilon_{\text{low}}, 1 + \epsilon_{\text{high}})$, where $\epsilon_{\text{low}} = 0.2$ and $\epsilon_{\text{high}} = 0.28$ by default. We relax either one end (*Relax-high* or *Relax-low*) or both ends (*Relax-both*). For example, Relax-high is implemented by ($\epsilon_{\text{low}} = 0.2, \epsilon_{\text{high}} = \infty$) for tokens with $\mu(y_t|s_t) < \alpha$.

As illustrated in Figure 6, the direction of clip relaxation plays a critical role in the training efficiency and stability. Relax-high can be viewed as an extreme variant of the Clip-Higher trick (Yu et al., 2025) applied only to low-probability tokens. While this approach maintains high entropy, it fails to yield significant gains in training efficiency. Conversely, Relax-low exhibits substantially faster initial learning[2]. However, this strategy eventually drops due to entropy collapse (Cui et al., 2025). Ultimately, we find that **Relax-both is the most effective strategy for achieving both efficient and stable training**, thereby validating the design of DPPO in relaxing both ends of the trust region.

## 7. Broader Evaluation

**Experimental Setting:** We conduct large-scale experiments to further validate our methods. We train on a filtered subset of DAPO-Math (Li et al., 2026), containing approximately 13k samples. Five model configurations (different base models and training techniques) are evaluated: (1) **MoE Base**: Qwen3-30B-A3B-Base (Yang et al., 2025); (2) **MoE Base**

w/ **R3**: Qwen3-30B-A3B-Base with rollout router replay (R3) (Ma et al., 2025); (3) **MoE Thinking**: Qwen3-30B-A3B; (4) **Dense Base**: Qwen3-8B-Base; (5) **MoE Base w/ LoRA**: Qwen3-30B-A3B-Base with LoRA (Hu et al., 2022). Baseline methods include **GRPO-ClipHigher**(Shao et al., 2024; Liu et al., 2025d; Yu et al., 2025) and **CISPO**(Chen et al., 2025; Khatri et al., 2025). All methods use the behavior policy ($\mu_{\theta'}$) instead of recomputed policy distribution ($\pi_{\theta'}$) to construct the trust region (i.e., for clipping or masking). We compare our proposed methods, **DPPO-Binary-KL** and **DPPO-Binary-TV**, against these baselines. More details are provided in Appendix F.

**Main Results.** We present online evaluation results on AIME24 and AIME25 (MAA, 2025) during RL training in the following figures: Figure 7 (MoE Base with and without R3) and Figure 8 (MoE Thinking and Dense Base). Results for MoE Base with LoRA are provided in Appendix G.2.

Our proposed method consistently demonstrates superior **stability** and **efficiency** across all five large-scale experiments. Specifically, DPPO optimizes rewards at a significantly faster speed than the GRPO-ClipHigher baseline and achieves better converged performance, providing empirical validation for the motivations discussed in Section 4.2. While all baseline methods frequently exhibit training instability or catastrophic collapse (e.g., CISPO in MoE Base without R3 and GRPO-ClipHigher in MoE Thinking), our approach maintains a remarkably stable training process.

Rollout router replay (R3) is widely considered a necessary technique for stabilizing RL training in MoE models (Ma et al., 2025; Zheng et al., 2025; Liu et al., 2025a). However, as illustrated in Figure 7, our DPPO variants (*without* R3) even consistently **outperform the R3-enhanced baselines**, which underscores the superior training efficiency and inherent stability of the DPPO framework. We provide additional detailed results and extended discussions in Appendix G.2.

**RLHF-style Alignment Tasks.** To evaluate DPPO beyond verifiable rewards, we further conduct RLHF experiments on open-ended alignment data. We compare GRPO with DPPO-Binary-TV in two settings: Gemma-2-9B-It (Team

---

[2]In contrast to the Clip-Higher intuition (Yu et al., 2025), we observe that "Clip-Lower" (relaxing $\epsilon_{\text{low}}$) for low-probability tokens is more vital for efficiency. This aligns with findings by Tajwar et al. (2024) regarding the role of negative gradients in accelerating preference learning.

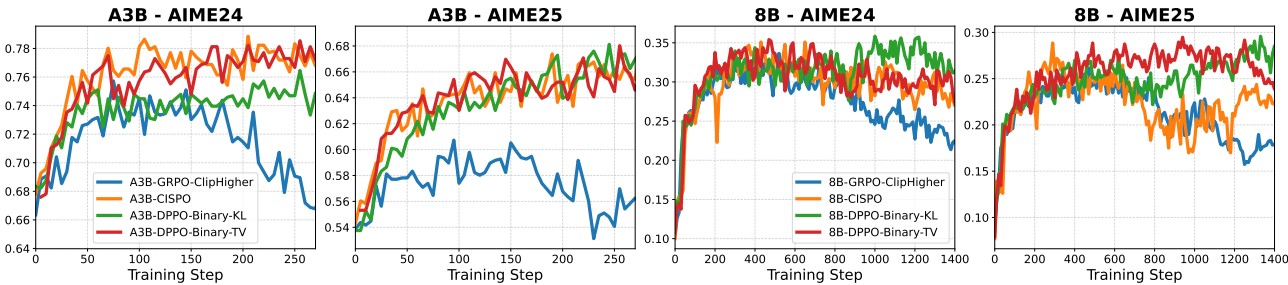

*Figure 8.* Evolution of AIME24 and AIME25 scores during RL training using Qwen3-30B-A3B (left) and Qwen3-8B-Base (right).

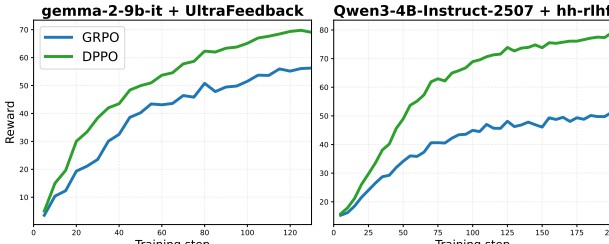

*Figure 9.* RLHF experiments with Skywork-Reward-Llama-3.1-8B as the reward model. DPPO-Binary-TV improves reward faster and reaches higher final reward than GRPO on both Gemma-2-9B-It with UltraFeedback and Qwen3-4B-Instruct-2507 with HH-RLHF.

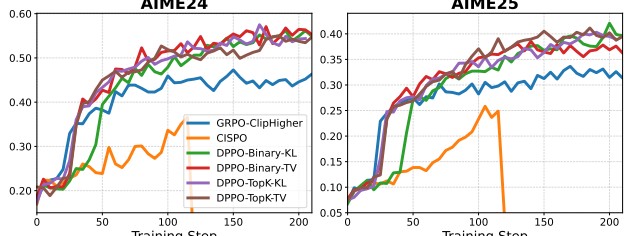

*Figure 10.* Evolution of AIME24 and AIME25 scores for baselines and DPPO with binary/Top-K (K=20) TV/KL approximation under the same setting as MoE Base w/o R3.

et al., 2024) trained on UltraFeedback (Cui et al., 2023), and Qwen3-4B-Instruct-2507 (Yang et al., 2025) trained on HH-RLHF (Bai et al., 2022). Both settings use Skywork-Reward-Llama-3.1-8B (Liu et al., 2024) as the reward model. As shown in Figure 9, DPPO improves the learned reward substantially faster and reaches higher final rewards in both settings. We further evaluate the fine-tuned Qwen3 models on AlpacaEval 2.0 in Appendix G.1, where DPPO achieves 80.90 length-controlled and 79.93 raw win rates, establishing a new SOTA on the AlpacaEval 2.0 community leaderboard. These results show that the effectiveness of DPPO also generalizes to noisy learned-reward optimization.

**Ablation on Divergence Approximation.** In the above scaling experiments, DPPO is implemented using the binary TV/KL approximation (Equations 13 and 14). To assess the impact of this simplification, we compare it against DPPO with the top-K (K=20) TV/KL (Equations 15 and 16) under the same setting as MoE Base. The results, presented in Figure 10, show that both approximations perform similarly and significantly outperform the baselines. This finding indicates that the easy-to-implement binary approximation is a sufficient and computationally efficient choice for scalable RL. We provide more detailed results in Appendix G.3.

**Generalization to Other Model Families and Tasks.** We also conduct experiments on Llama family models (Touvron et al., 2023; Wang et al., 2025b) and on general reasoning tasks (Liu et al., 2025e). The results, which are presented in Appendix G.5, show DPPO outperforms the baseline across most settings, highlighting its broad applicability.

**Hyperparameter Sensitivity.** We provide the hyperparameter sensitivity experiments in Appendix G.4.

## 8. Conclusion

In this work, we have presented a comprehensive rethinking of the trust region framework within the context of LLM fine-tuning. We derived policy improvement bounds specifically tailored to the finite-horizon, undiscounted setting of LLM generation, establishing a rigorous theoretical foundation for future trust-region research. Furthermore, through extensive empirical analysis, we investigated the trade-offs between training stability and efficiency, providing practical guidelines to optimize both.

Central to our contribution is the introduction of Divergence Proximal Policy Optimization (DPPO). We identified and addressed a critical structural flaw in PPO algorithm: it over-constrains updates to low-probability tokens while under-constraining potentially catastrophic shifts in high-probability tokens. This implicit bias results in a suboptimal training dynamic for the expansive, long-tailed vocabularies inherent to LLMs. By substituting heuristic ratio clipping with a more principled policy divergence, DPPO significantly enhances both efficiency and stability. To avoid computing an exact policy divergence, we introduced Binary and Top-K approximations, which capture essential divergence with negligible overhead. Our evaluations demonstrate that DPPO consistently outperforms existing methods like GRPO in both training efficiency and stability, offering a more robust foundation for the RL-based LLM fine-tuning.

## Impact Statement

This paper presents work whose goal is to advance the field of Machine Learning. There are many potential societal consequences of our work, none which we feel must be specifically highlighted here.

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

# A. Related Work

## A.1. Extended Connections to Existing Work

In this work, we identify a structural flaw in PPO's ratio-clipping mechanism within the LLM regime: it over-penalizes low-probability tokens and under-penalizes high-probability ones, thereby impairing training efficiency and stability. Our proposed DPPO addresses this issue by directly constraining the policy divergence. This methodology aligns with the insights of Wang et al. (2019; 2020), who observed similar exploration issues and proposed adaptive clipping based on KL divergence in traditional RL settings. However, in the context of LLMs, computing the exact divergence is prohibitive due to the huge memory footprint. To overcome this, we propose a binary divergence approximation, which empirically captures most of the benefits (see Appendix G.3). Furthermore, as demonstrated in Section 5 and Section 6, the challenges of training stability and efficiency are exacerbated in LLMs by their expansive vocabularies, because low-probability tokens form a non-trivial portion of the entire distribution due to the long-tailed nature (see Figure 1). Finally, the training-inference mismatch inherent to the LLM era introduces additional algorithmic complexities, as further detailed in Section 5.

## A.2. Training-inference Mismatch

Recent work has identified a key culprit for training instability: the *training-inference mismatch* ($\pi_\theta \neq \mu_\theta$), where the policy distribution used for gradient computation ($\pi_\theta$) diverges from the one used for data generation ($\mu_\theta$), even when using identical model parameters $\theta$ (Yao et al., 2025; Qi et al., 2025b; Liu et al., 2025b; Zheng et al., 2025). This discrepancy arises from numerical precision errors (Qi et al., 2025b) and subtle differences in implementation (Team et al., 2025a; He, 2025). As training progresses, this mismatch can be amplified if the RL algorithm cannot manage it appropriately, leading to catastrophic performance degradation (Qi et al., 2025b; Liu et al., 2025b).

Existing efforts to mitigate this issue primarily focus on correcting biased gradients through importance sampling. Building on this principle, techniques such as Truncated Importance Sampling (TIS) (Yao et al., 2025; Zheng et al., 2025) and Masked Importance Sampling (Liu et al., 2025b; Team et al., 2025b) have been introduced at both the token and sequence levels. However, as suggested by Qi et al. (2025b), these methods often fail to achieve a satisfactory balance between training efficiency and stability. In contrast, our DPPO algorithm significantly enhances both aspects compared to these existing approaches.

Another line of research attempts to resolve the mismatch issue through higher precision (Qi et al., 2025b) or rigorous engineering alignment (Team et al., 2025a; He, 2025; Zhang et al., 2025). While promising, these methods face limited applicability. For instance, aligning implementation details often requires specific training engines or model architectures, hindering broad adoption. Furthermore, in low-precision settings optimized for high-speed training, we must tolerate a significant training-inference mismatch. In such scenarios, a robust and fast algorithm like DPPO remains essential. Finally, our algorithmic design is orthogonal to these engineering-level optimizations and can be combined with them to achieve even greater performance gains.

# B. Trust Region in LLMs

## B.1. Proof of Performance Difference Identity

*Proof of Theorem 3.1.* We begin by expressing the difference in expected returns by its definition:

$$\mathcal{J}(\pi) - \mathcal{J}(\mu) = \mathbb{E}_{y \sim \pi}[R(y)] - \mathbb{E}_{y \sim \mu}[R(y)]$$
$$= \sum_y \big( \pi(y|x) - \mu(y|x) \big) R(y).$$

The core of the proof is to establish an identity for the difference in the probabilities of generating a sequence $y$, $\pi(y|x) - \mu(y|x)$. We use the following telescoping sum identity, which can be verified by expanding the terms:

$$\pi(y|x) - \mu(y|x) = \sum_{t=1}^{T} \left( \prod_{k=1}^{t-1} \mu(y_k|s_k) \right) \left( \pi(y_t|s_t) - \mu(y_t|s_t) \right) \left( \prod_{j=t+1}^{T} \pi(y_j|s_j) \right).$$

Substituting this identity into the expression for the performance difference yields:

$$\mathcal{J}(\pi) - \mathcal{J}(\mu) = \sum_y R(y) \sum_{t=1}^{T} \left( \prod_{k=1}^{t-1} \mu(y_k|s_k) \right) \left( \pi(y_t|s_t) - \mu(y_t|s_t) \right) \left( \prod_{j=t+1}^{T} \pi(y_j|s_j) \right)$$

$$= \sum_y \mu(y|x) R(y) \sum_{t=1}^{T} \left( \frac{\pi(y_t|s_t)}{\mu(y_t|s_t)} - 1 \right) \left( \prod_{j=t+1}^{T} \frac{\pi(y_j|s_j)}{\mu(y_j|s_j)} \right)$$

$$= \mathbb{E}_{y\sim\mu} \left[ R(y) \sum_{t=1}^{T} \left( \frac{\pi(y_t|s_t)}{\mu(y_t|s_t)} - 1 \right) \left( \prod_{j=t+1}^{T} \frac{\pi(y_j|s_j)}{\mu(y_j|s_j)} \right) \right]$$

$$= \mathbb{E}_{y\sim\mu} \left[ R(y) \sum_{t=1}^{|y|} \left( \frac{\pi(y_t|s_t)}{\mu(y_t|s_t)} - 1 \right) \right]$$

$$- \mathbb{E}_{y\sim\mu} \left[ R(y) \sum_{t=1}^{|y|} \left( \frac{\pi(y_t|s_t)}{\mu(y_t|s_t)} - 1 \right) \left( 1 - \prod_{j=t+1}^{T} \frac{\pi(y_j|s_j)}{\mu(y_j|s_j)} \right) \right].$$

By identifying the terms with the definitions in the theorem statement, we arrive at:

$$\mathcal{J}(\pi) - \mathcal{J}(\mu) = L'_\mu(\pi) - \Delta(\mu, \pi),$$

where

$$L'_\mu(\pi) = \mathbb{E}_{y\sim\mu} \left[ R(y) \sum_{t=1}^{|y|} \left( \frac{\pi(y_t|s_t)}{\mu(y_t|s_t)} - 1 \right) \right],$$

$$\Delta(\mu, \pi) = \mathbb{E}_{y\sim\mu} \left[ R(y) \sum_{t=1}^{|y|} \left( \frac{\pi(y_t|s_t)}{\mu(y_t|s_t)} - 1 \right) \left( 1 - \prod_{j=t+1}^{T} \frac{\pi(y_j|s_j)}{\mu(y_j|s_j)} \right) \right].$$

This completes the proof. $\qquad\square$

## B.2. Proof of Policy Improvement Bound: Max-Divergence Part

**Lemma B.1** (Bound on Sequence-Level TV Divergence). *Let $\mu$ and $\pi$ be two policies that generate sequences of length $N$. Let $\mu_N(\cdot|s_1)$ and $\pi_N(\cdot|s_1)$ denote the distributions over sequences $y = (y_1, \ldots, y_N)$. The total variation (TV) divergence between these sequence distributions is bounded by the sum of the expected single-step TV divergences:*

$$D_{\mathrm{TV}}\big(\mu_N(\cdot|s_1) \| \pi_N(\cdot|s_1)\big) \leq \sum_{t=1}^{N} \mathbb{E}_{s_t\sim\mu} \left[ D_{\mathrm{TV}}\big(\mu(\cdot|s_t) \| \pi(\cdot|s_t)\big) \right],$$

*where the expectation is over the state distribution induced by policy $\mu$.*

*Proof.* Let $P(y) = \mu_N(y|s_1)$ and $Q(y) = \pi_N(y|s_1)$.

$$2D_{\mathrm{TV}}(P\|Q) = \sum_y |P(y) - Q(y)| = \sum_y \left| \prod_{t=1}^{N} \mu(y_t|s_t) - \prod_{t=1}^{N} \pi(y_t|s_t) \right|.$$

We use the algebraic identity $a_1 \ldots a_N - b_1 \ldots b_N = \sum_{t=1}^{N} \left( \prod_{k=1}^{t-1} a_k \right) (a_t - b_t) \left( \prod_{j=t+1}^{N} b_j \right)$. Applying this to the policy

probabilities and then using the triangle inequality, we get:

$$2D_{\text{TV}}(P\|Q) \leq \sum_{y}\sum_{t=1}^{N}\left(\prod_{k=1}^{t-1}\mu(y_k|s_k)\right)|\mu(y_t|s_t)-\pi(y_t|s_t)|\left(\prod_{j=t+1}^{N}\pi(y_j|s_j)\right)$$

$$= \sum_{t=1}^{N}\sum_{y}\left(\prod_{k=1}^{t-1}\mu(y_k|s_k)\right)|\mu(y_t|s_t)-\pi(y_t|s_t)|\left(\prod_{j=t+1}^{N}\pi(y_j|s_j)\right).$$

For each term in the outer sum over $t$, we can sum over the variables $y_j$ for $j > t$. Since $\sum_{y_j}\pi(y_j|s_j) = 1$ for all $s_j$, the product of terms for $j > t$ sums to 1 when we integrate out $y_{t+1}, \ldots, y_N$. This leaves:

$$2D_{\text{TV}}(P\|Q) \leq \sum_{t=1}^{N}\sum_{y_1,\ldots,y_t}\left(\prod_{k=1}^{t-1}\mu(y_k|s_k)\right)|\mu(y_t|s_t)-\pi(y_t|s_t)|$$

$$= \sum_{t=1}^{N}\sum_{y_1,\ldots,y_{t-1}}\left(\prod_{k=1}^{t-1}\mu(y_k|s_k)\right)\sum_{y_t}|\mu(y_t|s_t)-\pi(y_t|s_t)|.$$

The inner sum is $2D_{\text{TV}}(\mu(\cdot|s_t)\|\pi(\cdot|s_t))$. The outer sum over $y_1, \ldots, y_{t-1}$ defines an expectation over states $s_t$ under policy $\mu$. Thus, we have:

$$2D_{\text{TV}}(P\|Q) \leq \sum_{t=1}^{N}\mathbb{E}_{s_t\sim\mu}\left[2D_{\text{TV}}\big(\mu(\cdot|s_t)\|\pi(\cdot|s_t)\big)\right].$$

Dividing by 2 yields the desired result. $\square$

*Proof of the max-divergence bound in Theorem 3.2.* From Lemma 3.1, we start with the exact performance difference identity:

$$\mathcal{J}(\pi) - \mathcal{J}(\mu) = L'_\mu(\pi) - \Delta(\mu, \pi).$$

For brevity, we define $y_{\leq t} = \{x, y_1, \ldots, y_t\}$ and $y_{>t} = \{y_{t+1}, y_{t+2}, \ldots\}$, then we can rewrite $\Delta(\mu, \pi)$ as:

$$\Delta(\mu, \pi) = \mathbb{E}_{y\sim\mu}\left[R(y)\sum_{t=1}^{|y|}\left(\frac{\pi(y_t|s_t)}{\mu(y_t|s_t)}-1\right)\left(1-\frac{\pi(y_{>t}|s_{t+1})}{\mu(y_{>t}|s_{t+1})}\right)\right].$$

Our goal is to find an upper bound for the error term $\Delta(\mu, \pi)$. We begin by bounding the reward by its maximum absolute value, $\xi = \max_y |R(y)|$.

$$\Delta(\mu, \pi) \leq \xi \cdot \mathbb{E}_{y\sim\mu}\left[\sum_{t=1}^{T}\left|\frac{\pi(y_t|s_t)}{\mu(y_t|s_t)}-1\right|\cdot\left|1-\frac{\pi(y_{>t}|s_{t+1})}{\mu(y_{>t}|s_{t+1})}\right|\right]$$

$$= \xi \cdot \sum_{t=1}^{T}\mathbb{E}_{y_{\leq t}\sim\mu}\left[\left|\frac{\pi(y_t|s_t)}{\mu(y_t|s_t)}-1\right|\cdot\mathbb{E}_{y_{>t}\sim\mu(\cdot|s_{t+1})}\left[\left|1-\frac{\pi(y_{>t}|s_{t+1})}{\mu(y_{>t}|s_{t+1})}\right|\right]\right]. \quad (17)$$

The inner expectation is exactly twice the TV divergence between the distributions over future trajectories:

$$\mathbb{E}_{y_{>t}\sim\mu(\cdot|s_{t+1})}\left[\left|1-\frac{\pi(y_{>t}|s_{t+1})}{\mu(y_{>t}|s_{t+1})}\right|\right] = 2D_{\text{TV}}\big(\mu_{>t}(\cdot|s_{t+1})\|\pi_{>t}(\cdot|s_{t+1})\big).$$

Using Theorem B.1 on this sequence-level TV divergence (for a sequence of length $T - t$), we get:

$$D_{\text{TV}}\big(\mu_{>t}(\cdot|s_{t+1})\|\pi_{>t}(\cdot|s_{t+1})\big) \leq \sum_{k=t+1}^{T}\mathbb{E}_{s_k\sim\mu(\cdot|s_{t+1})}\left[D_{\text{TV}}\big(\mu(\cdot|s_k)\|\pi(\cdot|s_k)\big)\right].$$

We bound each term in the sum by the maximum single-step TV divergence, $D_{\mathrm{TV}}^{\max}(\mu \,\|\, \pi) = \max_s D_{\mathrm{TV}}(\mu(\cdot|s)\|\,\pi(\cdot|s))$, which gives:

$$D_{\mathrm{TV}}\big(\mu_{>t}(\cdot|s_{t+1})\|\pi_{>t}(\cdot|s_{t+1})\big) \le \sum_{k=t+1}^{T} D_{\mathrm{TV}}^{\max}(\mu \,\|\, \pi) = (T-t)D_{\mathrm{TV}}^{\max}(\mu \,\|\, \pi).$$

Substituting this back into the bound for $\Delta(\mu, \pi)$:

$$
\begin{aligned}
\Delta(\mu, \pi) &\le \xi \cdot \sum_{t=1}^{T} \mathbb{E}_{y_{\le t}\sim\mu}\left[\left|\frac{\pi(y_t|s_t)}{\mu(y_t|s_t)} - 1\right| \cdot 2(T-t)D_{\mathrm{TV}}^{\max}(\mu \,\|\, \pi)\right] \\
&= 2\xi \cdot D_{\mathrm{TV}}^{\max}(\mu \,\|\, \pi) \sum_{t=1}^{T}(T-t)\mathbb{E}_{s_t\sim\mu}\left[\sum_{y_t}\mu(y_t|s_t)\left|\frac{\pi(y_t|s_t)}{\mu(y_t|s_t)} - 1\right|\right] \\
&= 2\xi \cdot D_{\mathrm{TV}}^{\max}(\mu \,\|\, \pi) \sum_{t=1}^{T}(T-t)\mathbb{E}_{s_t\sim\mu}\left[2D_{\mathrm{TV}}(\mu(\cdot|s_t)\|\,\pi(\cdot|s_t))\right] \\
&\le 2\xi \cdot D_{\mathrm{TV}}^{\max}(\mu \,\|\, \pi) \sum_{t=1}^{T}(T-t)\cdot\mathbb{E}_{s_t\sim\mu}\left[2D_{\mathrm{TV}}^{\max}(\mu \,\|\, \pi)\right] \\
&= 4\xi \cdot D_{\mathrm{TV}}^{\max}(\mu \,\|\, \pi)^2 \sum_{t=1}^{T}(T-t) \\
&= 2\xi T(T-1) \cdot D_{\mathrm{TV}}^{\max}(\mu \,\|\, \pi)^2.
\end{aligned}
\tag{18}
$$

Substituting this into the performance difference identity gives the max-divergence bound in Equation (7):

$$\mathcal{J}(\pi) - \mathcal{J}(\mu) \ge L'_\mu(\pi) - 2\xi T(T-1) \cdot D_{\mathrm{TV}}^{\max}(\mu \,\|\, \pi)^2.$$

This completes the proof. $\qquad\square$

### B.3. Proof of Policy Improvement Bound: Average-Divergence Part

We now prove the linear average-divergence bound in Equation (8), which is the second part of Theorem 3.2. Compared with the quadratic max-divergence bound in Equation (7), this form avoids a $T^2$ dependence and is therefore more suitable for long LLM responses. The key step is to use the fact that total variation is always bounded by one, i.e., $D_{\mathrm{TV}}(P\|Q) \le 1$, instead of upper-bounding the future-trajectory divergence by $(T-t)D_{\mathrm{TV}}^{\max}(\mu \,\|\, \pi)$.

We begin from the intermediate step in Equation (17):

$$\Delta(\mu, \pi) \le \xi \cdot \sum_{t=1}^{T} \mathbb{E}_{y_{\le t}\sim\mu}\left[\left|\frac{\pi(y_t|s_t)}{\mu(y_t|s_t)} - 1\right| \cdot \mathbb{E}_{y_{>t}\sim\mu(\cdot|s_{t+1})}\left[\left|1 - \frac{\pi(y_{>t}|s_{t+1})}{\mu(y_{>t}|s_{t+1})}\right|\right]\right].$$

The inner expectation is exactly twice the TV divergence between the future trajectory distributions, $2D_{\mathrm{TV}}\big(\mu_{>t}(\cdot|s_{t+1})\|\pi_{>t}(\cdot|s_{t+1})\big)$. Instead of bounding this term with $2(T-t)D_{\mathrm{TV}}^{\max}(\mu \,\|\, \pi)$, we now apply the simple

upper bound of 2:

$$\Delta(\mu,\pi) \leq \xi \cdot \sum_{t=1}^{T} \mathbb{E}_{y_{\leq t}\sim\mu}\left[\left|\frac{\pi(y_t|s_t)}{\mu(y_t|s_t)} - 1\right| \cdot 2D_{\mathrm{TV}}\big(\mu_{>t}(\cdot|s_{t+1})\|\pi_{>t}(\cdot|s_{t+1})\big)\right]$$

$$\leq 2\xi \cdot \sum_{t=1}^{T} \mathbb{E}_{y_{\leq t}\sim\mu}\left[\left|\frac{\pi(y_t|s_t)}{\mu(y_t|s_t)} - 1\right|\right] \tag{19}$$

$$= 2\xi \cdot \sum_{t=1}^{T} \mathbb{E}_{s_t\sim\rho_t^\mu}\mathbb{E}_{y_t\sim\mu(\cdot|s_t)}\left[\left|\frac{\pi(y_t|s_t)}{\mu(y_t|s_t)} - 1\right|\right]$$

$$= 2\xi \cdot \sum_{t=1}^{T} \mathbb{E}_{s_t\sim\rho_t^\mu}\left[2D_{\mathrm{TV}}(\mu(\cdot|s_t)\|\pi(\cdot|s_t))\right]$$

$$= 4\xi \cdot \mathbb{E}_{y\sim\mu}\left[\sum_{t=1}^{|y|} D_{\mathrm{TV}}(\mu(\cdot|s_t)\|\pi(\cdot|s_t))\right]. \tag{20}$$

Substituting Equation (20) into the performance difference identity proves the average-divergence part of Theorem 3.2:

$$\mathcal{J}(\pi) - \mathcal{J}(\mu) \geq L'_\mu(\pi) - 4\xi \cdot \mathbb{E}_{y\sim\mu}\left[\sum_{t=1}^{|y|} D_{\mathrm{TV}}(\mu(\cdot|s_t)\|\pi(\cdot|s_t))\right].$$

Equivalently, using the notation in Theorem 3.2, this is

$$\mathcal{J}(\pi) - \mathcal{J}(\mu) \geq L'_\mu(\pi) - 4\xi\bar{D}_{\mathrm{TV}}(\mu,\pi).$$

Since both bounds hold simultaneously, an immediate corollary is the following composite guarantee:

$$\mathcal{J}(\pi) - \mathcal{J}(\mu) \geq L'_\mu(\pi) - \Delta(\mu,\pi)$$

$$\geq L'_\mu(\pi) - \min\left(2\xi T(T-1)\cdot D_{\mathrm{TV}}^{\max 2}, 4\xi\cdot\mathbb{E}_{y\sim\mu}\left[\sum_{t=1}^{|y|} D_{\mathrm{TV}}(\mu(\cdot|s_t)\|\pi(\cdot|s_t))\right]\right).$$

This composite bound leverages the quadratic bound for infinitesimal updates and the linear bound for larger updates or longer horizons.

### B.4. Comparing Surrogate Objectives with Classical RL

At first glance, the surrogate objective for the LLM regime in Equation (5) appears distinct from the classical RL surrogate in Equation (3). The former is an expectation over full trajectories $y$ weighted by the reward $R(y)$, while the latter is an expectation over state-action pairs $(s,a)$ weighted by the advantage $A^\mu(s,a)$. However, we will now show that their gradients with respect to the policy parameters $\theta$ are fundamentally analogous, confirming that our LLM-specific formulation is a valid adaptation of the standard policy gradient theorem.

Let the policy $\pi$ be parameterized by $\theta$. We will use the identity $\nabla_\theta\pi_\theta(a|s) = \pi_\theta(a|s)\nabla_\theta\log\pi_\theta(a|s)$.

**Gradient of the Classical Surrogate Objective.** We begin with the classical surrogate objective from Equation (3):

$$L_\mu(\pi_\theta) = \frac{1}{1-\gamma}\mathbb{E}_{s\sim\rho^\mu,\, a\sim\mu(a|s)}\left[\frac{\pi_\theta(a|s)}{\mu(a|s)}A^\mu(s,a)\right].$$

Taking the gradient with respect to $\theta$ and moving it inside the expectation, we get:

$$\nabla_\theta L_\mu(\pi_\theta) = \frac{1}{1-\gamma}\mathbb{E}_{s\sim\rho^\mu,\, a\sim\mu(a|s)}\left[\frac{\nabla_\theta\,\pi_\theta(a|s)}{\mu(a|s)}A^\mu(s,a)\right]$$

$$= \frac{1}{1-\gamma}\mathbb{E}_{s\sim\rho^\mu,\, a\sim\mu(a|s)}\left[\frac{\pi_\theta(a|s)}{\mu(a|s)}\nabla_\theta\log\pi_\theta(a|s)A^\mu(s,a)\right]. \tag{21}$$

**Gradient of the LLM Surrogate Objective.** Next, we consider our LLM-specific surrogate from Equation (5):

$$L'_\mu(\pi_\theta) = \mathbb{E}_{y \sim \mu} \left[ R(y) \sum_{t=1}^{|y|} \left( \frac{\pi_\theta(y_t|s_t)}{\mu(y_t|s_t)} - 1 \right) \right].$$

Taking the gradient with respect to $\theta$ and noting that the $-1$ term has a zero gradient:

$$\nabla_\theta L'_\mu(\pi_\theta) = \mathbb{E}_{y \sim \mu} \left[ R(y) \sum_{t=1}^{|y|} \frac{\nabla_\theta \pi_\theta(y_t|s_t)}{\mu(y_t|s_t)} \right]$$

$$= \mathbb{E}_{y \sim \mu} \left[ \sum_{t=1}^{|y|} \frac{\pi_\theta(y_t|s_t)}{\mu(y_t|s_t)} \nabla_\theta \log \pi_\theta(y_t|s_t) R(y) \right].$$

If we define a sequence-level advantage as $A^\mu(s_t, y_t) = R(y) - V(x)$, where $V(x)$ is a baseline value function for the prompt, the gradient becomes:

$$\nabla_\theta L'_\mu(\pi_\theta) = \mathbb{E}_{y \sim \mu} \left[ \sum_{t=1}^{|y|} \frac{\pi_\theta(y_t|s_t)}{\mu(y_t|s_t)} \nabla_\theta \log \pi_\theta(y_t|s_t) A^\mu(s_t, y_t) \right]. \tag{22}$$

This form is directly analogous to the classical policy gradient in Equation (21), where the sum over timesteps in a trajectory replaces the expectation over the state distribution $\rho^\mu$. Thus, our LLM surrogate objective is a theoretically sound adaptation of the classical trust region framework to the undiscounted, sequence-reward setting.

## C. Approximations as Lower Bounds of True Divergence

In this section, we provide a formal justification for our Binary and Top-K divergence approximations. We demonstrate that both are principled lower bounds on the true divergence and explicitly state the conditions under which these approximations become exact.

Let $\mathcal{C} = \{C_1, \ldots, C_m\}$ be any partition of the vocabulary $\mathcal{A}$. Our Binary and Top-K approximations correspond to specific choices of this partition. We will show that the divergence computed on the partitioned space is a lower bound on the true divergence.

### C.1. Total Variation Divergence

The true TV divergence is $D_{\text{TV}}(\mu \| \pi) = \frac{1}{2} \sum_{a \in \mathcal{A}} |\mu(a|s_t) - \pi(a|s_t)|$. The divergence on a partitioned space $\mathcal{C}$ is $D^{\mathcal{C}}_{\text{TV}} = \frac{1}{2} \sum_{j=1}^{m} |\mu(C_j|s_t) - \pi(C_j|s_t)|$.

**Proof of Lower Bound.** By definition, $|\mu(C_j|s_t) - \pi(C_j|s_t)| = |\sum_{a \in C_j} (\mu(a|s_t) - \pi(a|s_t))|$. The triangle inequality states that the absolute value of a sum is less than or equal to the sum of the absolute values. Applying this, we get $|\sum_{a \in C_j} (\mu(a|s_t) - \pi(a|s_t))| \leq \sum_{a \in C_j} |\mu(a|s_t) - \pi(a|s_t)|$. Summing over all partitions $j$:

$$D^{\mathcal{C}}_{\text{TV}} = \frac{1}{2} \sum_{j=1}^{m} \left| \sum_{a \in C_j} (\mu(a|s_t) - \pi(a|s_t)) \right|$$

$$\leq \frac{1}{2} \sum_{j=1}^{m} \sum_{a \in C_j} |\mu(a|s_t) - \pi(a|s_t)|$$

$$= D_{\text{TV}}(\mu \| \pi).$$

Thus, $D_{\text{TV}}(\mu \| \pi) \geq D^{\mathcal{C}}_{\text{TV}}$. This holds for both Binary and Top-K partitions.

**Analysis of the Approximation Gap.** The gap between the true and approximated TV divergence is the sum of the gaps within each partition. For any partition $C_j$, the gap is $\frac{1}{2} \left( \sum_{a \in C_j} |\mu(a|s_t) - \pi(a|s_t)| - \left| \sum_{a \in C_j} (\mu(a|s_t) - \pi(a|s_t)) \right| \right)$.

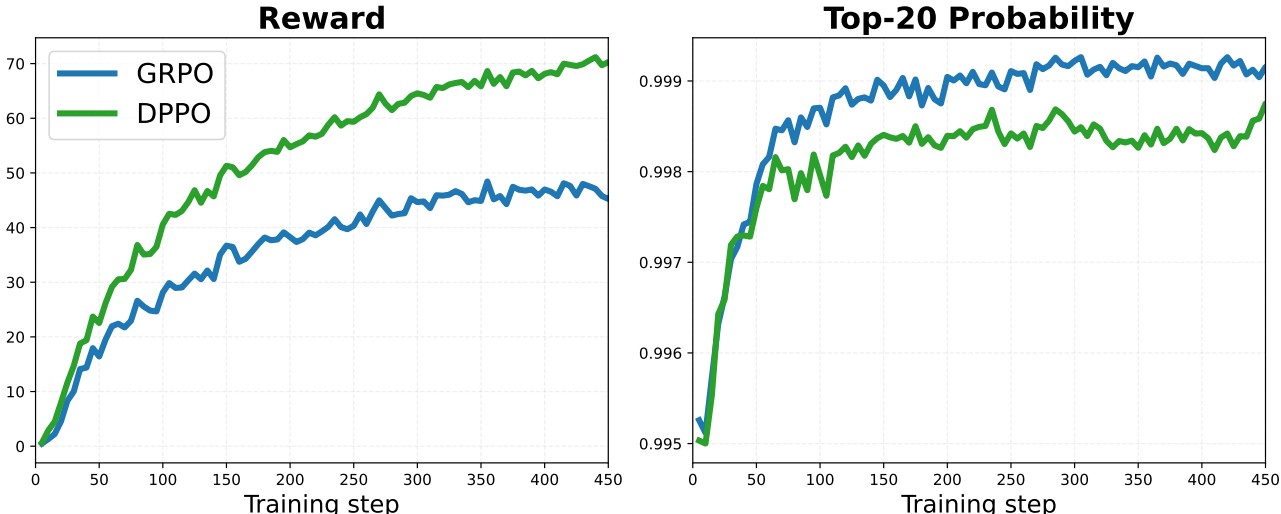

*Figure 11.* RLHF training curves and top-20 probability mass over training. The top-20 tokens cover nearly all probability mass throughout training, making the Top-K tail and the resulting TV approximation gap small.

This gap is bounded by the total probability mass of the partition:

$$\text{Gap}(C_j) \leq \frac{1}{2} \sum_{a \in C_j} (\mu(a|s_t) + \pi(a|s_t)) = \frac{1}{2}(\mu(C_j|s_t) + \pi(C_j|s_t)).$$

For the Top-K approximation, the only partition with a potential gap is the "other" category, which contains the tail of the distribution. The total probability mass of this tail, $\mu(C_{\text{other}}|s_t)$, is typically very small. Therefore, the approximation gap is also small, justifying Top-K TV as a high-fidelity approximation.

We further validate that this tail-mass bound remains small across training. In an RLHF run, we track the average probability mass covered by the top-20 tokens, matching the $K = 20$ Top-K approximation used in our experiments. As shown in Figure 11, the top-20 mass is already 99.4% at the beginning of training and increases to 99.9% after 100 training steps. Equivalently, the "other" category contains only 0.6% of the mass initially and about 0.1% after the policy sharpens, so the Top-K TV approximation gap remains tiny and empirically improves during training rather than degrading.

**Equality Condition.** Equality $D_{\text{TV}} = D_{\text{TV}}^{\mathcal{C}}$ holds if the gap is zero for all partitions. This occurs when $\mu(a|s_t) - \pi(a|s_t)$ has the same sign for all tokens $a$ within each partition $C_j$.

### C.2. KL Divergence

The true KL divergence is $D_{\text{KL}}(\mu \| \pi) = \sum_{a \in \mathcal{A}} \mu(a|s_t) \log \frac{\mu(a|s_t)}{\pi(a|s_t)}$. The divergence on the partitioned space is $D_{\text{KL}}^{\mathcal{C}} = \sum_{j=1}^{m} \mu(C_j|s_t) \log \frac{\mu(C_j|s_t)}{\pi(C_j|s_t)}$.

**Proof of Lower Bound.** The proof relies on the log-sum inequality, which states that for any two sets of non-negative numbers $\{x_1, \ldots, x_n\}$ and $\{y_1, \ldots, y_n\}$:

$$\sum_{i=1}^{n} x_i \log \frac{x_i}{y_i} \geq \left( \sum_{i=1}^{n} x_i \right) \log \frac{\sum_{i=1}^{n} x_i}{\sum_{i=1}^{n} y_i}.$$

We apply this inequality to each partition $C_j$ in our vocabulary, setting $x_a = \mu(a|s_t)$ and $y_a = \pi(a|s_t)$:

$$\sum_{a \in C_j} \mu(a|s_t) \log \frac{\mu(a|s_t)}{\pi(a|s_t)} \geq \left( \sum_{a \in C_j} \mu(a|s_t) \right) \log \frac{\sum_{a \in C_j} \mu(a|s_t)}{\sum_{a \in C_j} \pi(a|s_t)}$$

$$= \mu(C_j|s_t) \log \frac{\mu(C_j|s_t)}{\pi(C_j|s_t)}.$$

Summing over all partitions $j$ gives the desired result:

$$D_{\text{KL}}(\mu \| \pi) = \sum_{j=1}^{m} \sum_{a \in C_j} \mu(a|s_t) \log \frac{\mu(a|s_t)}{\pi(a|s_t)}$$

$$\geq \sum_{j=1}^{m} \mu(C_j|s_t) \log \frac{\mu(C_j|s_t)}{\pi(C_j|s_t)} = D_{\text{KL}}^{\mathcal{C}}.$$

**Equality Condition.** The log-sum inequality holds with equality if and only if the ratio $\frac{x_i}{y_i}$ is constant for all $i$. In our context, this means that for each partition $C_j$, the ratio $\frac{\mu(a|s_t)}{\pi(a|s_t)}$ must be constant for all tokens $a \in C_j$. For both Binary and Top-K approximations, this implies the policy update must scale the probabilities of all tokens within the "other" category by a uniform factor.

## D. More Details for Stability Analysis

Our experimental setup strictly follows the sanity test established in Qi et al. (2025b). Each policy iteration begins by sampling a batch of 64 questions. For each question, we generate 8 responses (rollouts) using a maximum context length of 8,000. The collected data is then used to perform 4 gradient steps. All experiments are conducted using the VeRL framework (Sheng et al., 2024) together with the ODC optimization (Wan et al., 2026), and models are trained in BFloat16 precision to better expose potential numerical instabilities between algorithms. For the evaluation on AIME, we sample 32 responses for each test question to ensure a robust assessment.

### D.1. Algorithmic Details for Stability Analysis

In this section, we provide the specific policy gradient formulations for each algorithm evaluated in our stability analysis (Section 5). To facilitate a direct comparison, we show how each algorithm's gradient update can be interpreted through the lens of a single, unified framework.

**A Unified Policy Gradient Formulation.** The policy gradient for the algorithms we tested can be generalized into the following form, where the gradient of the objective $L(\theta)$ is expressed as:

$$\nabla_\theta L(\theta) = \mathbb{E}_{y \sim \mu_{\theta'}} \left[ \sum_{t=1}^{|y|} M_t \cdot \min \left( \frac{\pi_\theta(y_t|s_t)}{\mu_{\theta'}(y_t|s_t)}, C \right) \cdot \hat{A}_t \cdot \nabla_\theta \log \pi_\theta(y_t|s_t) \right]. \tag{23}$$

In this formulation, $\hat{A}_t$ is the advantage, estimated following the GRPO method but without standard deviation normalization (Shao et al., 2024; Liu et al., 2025d). The algorithms differ primarily in their definition of the binary mask $M_t$ and the clipping bound $C$.

- For **PG-IS**, we have $M_t = 1$ and $C = \infty$.

- For **PG-TIS (CISPO)**, we have $M_t = 1$ and $C = 3$.

- For **GRPO**, the mask $M_t$ is the PPO-style clipping mask, and $C = \infty$.

- For **MiniRL**, the mask $M_t$ is also a PPO-style clipping mask but is conditioned on a recomputed policy ratio. For this algorithm, $C = \infty$.

- For **MiniRL-TIS**, the mask $M_t$ is the same as in MiniRL, but with $C = 3$.

- For **DPPO (Ours)**, the mask $M_t$ is conditioned on the policy divergence, and $C = \infty$.

**Mask Definitions.** The specific forms of the masks are as follows:

- For **GRPO**, the mask uses the rollout ratio $r_t = \frac{\pi_\theta(y_t|s_t)}{\mu_{\theta'}(y_t|s_t)}$ and experimental hyperparameters $\epsilon_{\text{high}} = 0.28, \epsilon_{\text{low}} = 0.2$:

$$M_t = \begin{cases} 0, & \text{if } (\hat{A}_t > 0 \text{ and } r_t > 1 + \epsilon_{\text{high}}) \text{ or } (\hat{A}_t < 0 \text{ and } r_t < 1 - \epsilon_{\text{low}}) \\ 1, & \text{otherwise.} \end{cases}$$

- For **MiniRL** and **MiniRL-TIS**, the mask is structurally identical to GRPO's and uses the same hyperparameters, but it is conditioned on the recomputed ratio $r'_t = \frac{\pi_\theta(y_t|s_t)}{\pi_{\theta'}(y_t|s_t)}$:

$$M_t = \begin{cases} 0, & \text{if } (\hat{A}_t > 0 \text{ and } r'_t > 1 + \epsilon_{\text{high}}) \text{ or } (\hat{A}_t < 0 \text{ and } r'_t < 1 - \epsilon_{\text{low}}) \\ 1, & \text{otherwise.} \end{cases}$$

- For **DPPO**, our mask is conditioned on the policy divergence $D_t$:

$$M_t = \begin{cases} 0, & \text{if } (\hat{A}_t > 0 \text{ and } r_t > 1 \text{ and } D_t > \delta) \text{ or } (\hat{A}_t < 0 \text{ and } r_t < 1 \text{ and } D_t > \delta) \\ 1, & \text{otherwise.} \end{cases}$$

In our experiments, we set the divergence threshold $\delta = 0.15$ for TV divergence and $\delta = 0.05$ for KL divergence.

## E. Characterizing Clipped Tokens

To understand the practical consequences of ratio clipping, we analyzed which tokens are most frequently penalized by a PPO-style algorithm. We trained a Qwen3-4B-Base model on the DAPO dataset with GRPO and, at training step 50, collected two sets of tokens:

- **Clipped Positive Tokens:** From samples with $\hat{A}_t > 0$, tokens whose updates were blocked due to a high ratio ($r_t > 1.28$).

- **Clipped Negative Tokens:** From samples with $\hat{A}_t < 0$, tokens whose updates were blocked due to a low ratio ($r_t < 0.8$).

---

The 50 most frequently clipped tokens from **positively-rewarded** samples.

```
' the', ' \\(', '1', 'Let', ' in', ' ', ',', 'We', ' +', ' \\', ' numbers',
':\n\n', 'Wait', '4', '6', ' Identify', '(', 'Next', ' from', ')', ' k', ' -',
'Since', ' solve', '\\[', ' how', ' ->', ' to', ' are', 'Sub', 'I', '):\n', '
\n\n', ' spiral', ' Instead', ' this', 'If', 'div', ' Conditions', ' vector', '
have', ' =', ' feasible', 'Or', ' inconsistency', ' express', '_{', ' increase', '
exact', ' consider'
```

---

The 50 most frequently clipped tokens from **negatively-rewarded** samples.

```
' \\(', ' the', ',', ' a', ' \\', ' ', '2', '1', ':\n\n', '0', '3', ' and', ' (', '
that', '-', ' to', '5', ' of', 'However', '\\', ' is', ' =', '4', ' in', ' for', '
all', ' we', 'We', ')', '.\n\n', ' our', '.', ':\n', ' but', ' with', 'So', '
both', 'From', ' Let', ' this', 'Thus', 'Wait', ' if', ' -', ' +', '^', ' only', '
at', 'Since', ' integer'
```

---

The 50 most frequent tokens in each category reveal a striking pattern. Far from being random noise, the clipped tokens are often critical for task performance. The lists for both positive and negative samples are dominated by two key categories:

1. **Numerical and Mathematical Tokens:** A significant portion of the clipped tokens are numbers (e.g., '1', '4') and mathematical symbols (e.g., '+', '=', 'div').

2. **Reasoning and Structural Words:** The list also includes many words essential for logical exposition, such as 'Wait', 'Next', 'Thus', and 'Since'.

These findings highlight a fundamental flaw in ratio-based clipping. For positive samples, it blocks beneficial updates to tokens that are integral to constructing correct solutions. For negative samples, it blocks the necessary suppression of these same tokens when they are part of an incorrect reasoning path. By systematically interfering with the learning signal for these high-utility tokens, the algorithm inadvertently slows learning, stifles exploration, and hinders the model's ability to refine its problem-solving capabilities.

# F. More Details for Broader Evaluation

In this section, we provide detailed training and evaluation settings of the **scaling experiments** in Section 7.

**Training Settings.** We conduct experiments using the VeRL framework (Sheng et al., 2024) on NVIDIA H Series GPUs. All methods follow the hyperparameter configurations detailed in Table 1. Rollout router replay (R3) (Ma et al., 2025) records the routed experts used in the inference engine and replays them in the training engine, which mitigates the training-inference mismatch and stabilizes RL training for MoE models. We only use R3 in the MoE Base w/ R3 experiment and do not use it in all other experiments. For experiments that utilize LoRA, as suggested by Schulman & Thinking Machines Lab (2025), we employ a larger learning rate of $1 \times 10^{-5}$. For the MoE Base w/ LoRA experiment, we set `lora_rank=32` and `lora_alpha=64`.

As suggested in Section 5.2, for all methods, we use the behavior policy ($\mu_{\theta'}$) instead of recomputed policy distribution ($\pi_{\theta'}$) to construct the trust region (i.e., for clipping or masking). Under the unified policy gradient formulation (Equation 23), the method-specific hyperparameters ($C = 5$ by default) are configured as follows:

- For **GRPO-ClipHigher**, we have

$$M_t = \begin{cases} 0, & \text{if } (\hat{A}_t > 0 \text{ and } r_t > 1 + \epsilon_{\text{high}}) \text{ or } (\hat{A}_t < 0 \text{ and } r_t < 1 - \epsilon_{\text{low}}) \\ 1, & \text{otherwise.} \end{cases}$$

  where $\epsilon_{\text{high}} = 0.27$ and $\epsilon_{\text{low}} = 0.2$, which follows the hyperparameters used in Zheng et al. (2025).

- For **CISPO**, we have $M_t = 1$.

- For **DPPO-Binary-KL** and **DPPO-Binary-TV**, we have

$$M_t = \begin{cases} 0, & \text{if } (\hat{A}_t > 0 \text{ and } r_t > 1 \text{ and } D_t > \delta) \text{ or } (\hat{A}_t < 0 \text{ and } r_t < 1 \text{ and } D_t > \delta) \\ 1, & \text{otherwise.} \end{cases}$$

  where $D_t$ is binary approximation of KL or TV as defined in Section 4.4. For **DPPO-Binary-KL**, $\delta = 0.05$ for all scaling experiments. For **DPPO-Binary-TV**, we use $\delta = 0.15$ for MoE Base w/ LoRA experiment and $\delta = 0.2$ for all other scaling experiments.

*Table 1.* Detailed RL training hyperparameters of scaling experiments.

| Hyperparameters | MoE Base | MoE Base w/ R3 | MoE Thinking | Dense Base | MoE Base w/ LoRA |
|---|---|---|---|---|---|
| `max_prompt_length` | 1024 | 1024 | 1024 | 1024 | 1024 |
| `max_response_length` | 16384 | 16384 | 16384 | 8000 | 8000 |
| `train_batch_size` | 256 | 256 | 256 | 128 | 128 |
| `ppo_mini_batch_size` | 32 | 32 | 32 | 32 | 16 |
| `optim.lr` | 1e-6 | 1e-6 | 1e-6 | 1e-6 | 1e-5 |
| `rollout.temperature` | 1.0 | 1.0 | 1.0 | 1.0 | 1.0 |
| `rollout.n` | 16 | 16 | 16 | 8 | 8 |
| **Detailed Results** | Figure 12 | Figure 13 | Figure 14 | Figure 15 | Figure 16 |

**Evaluation Settings.** We perform online evaluation for each method and experimental configuration, monitoring AIME24 and AIME25 scores throughout RL training. Evaluations are conducted every 5 training steps for MoE Base, MoE Base w/ R3, and MoE Thinking, and every 10 steps for Dense Base and MoE Base w/ LoRA.

Across all scaling experiments, we use consistent sampling parameters: `temperature=0.7`, `top_p=0.95`, and `n=32`. The `n=32` setting indicates that each question from AIME24 and AIME25 is sampled 32 times, and we report the average scores. The `max_response_length` remains identical to that used during training rollouts.

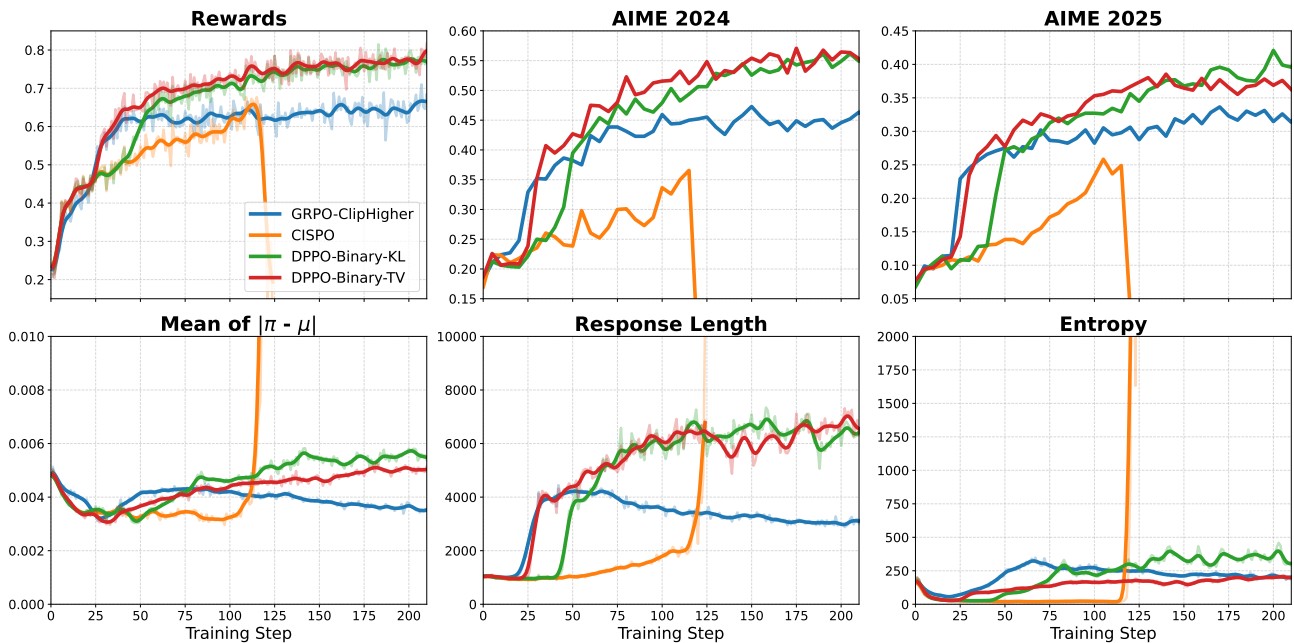

*Figure 12.* Evolution of metrics for **MoE Base w/o R3** experiment (based on Qwen3-30B-A3B-Base, without rollout router replay).

## G. More Empirical Results

### G.1. AlpacaEval 2.0 Evaluation for RLHF

To complement the RLHF experiments, we also evaluate the Qwen3-4B-Instruct-2507 model fine-tuned on UltraFeedback with Skywork-Reward-Llama-3.1-8B on AlpacaEval 2.0. In this run, DPPO also obtains higher learned reward than GRPO at step 150 (51.32 vs. 36.70) and step 450 (70.27 vs. 45.24). We report the checkpoint at step 150 for AlpacaEval, since later checkpoints obtain lower AlpacaEval scores despite higher reward, indicating reward hacking.

*Table 2.* AlpacaEval 2.0 results for RLHF fine-tuning on Qwen3-4B-Instruct-2507 with UltraFeedback. We report length-controlled win rate (LC-WR), raw win rate (WR), and average response length.

| Model | LC-WR | WR | Avg. Length |
|---|---|---|---|
| Initial model | 59.39 | 69.38 | 3147 |
| GRPO fine-tuning | 77.05 | 72.20 | 1756 |
| DPPO fine-tuning | **80.90** | **79.93** | 2003 |

As shown in Table 2, DPPO achieves the highest length-controlled and raw win rates among the compared checkpoints, establishing a new SOTA on the AlpacaEval 2.0 community leaderboard. This indicates that the reward improvement from DPPO transfers to an external open-ended alignment benchmark, rather than only increasing the training reward model score.

### G.2. Extended Main Results

In addition to the results provided in Section 7, here we provide more detailed results of the five scaling experiments: Figure 12 for MoE Base w/o R3, Figure 13 for MoE Base w/ R3, Figure 14 for MoE Thinking, Figure 15 for Dense Base, Figure 16 for MoE Base w/ LoRA. We record the following metrics throughout the RL training: training rewards (denoted as "**Rewards**"), **AIME 2024** Avg@32 scores, **AIME 2025** Avg@32 scores, mean of $|\mu_{\theta'} - \pi_{\theta'}|$ (denoted as "**Mean of** $|\pi - \mu|$"), mean of the response length (denoted as "**Response Length**"), and mean of token entropy (denoted as "**Entropy**"). For clearer visualization, all metrics except AIME24 and AIME25 are smoothed using a Gaussian filter with standard deviation $\sigma = 2$. The original unsmoothed curves are shown in the background as shaded regions.

Overall, across the five experiments, our method DPPO demonstrates consistent and robust improvements in training rewards,

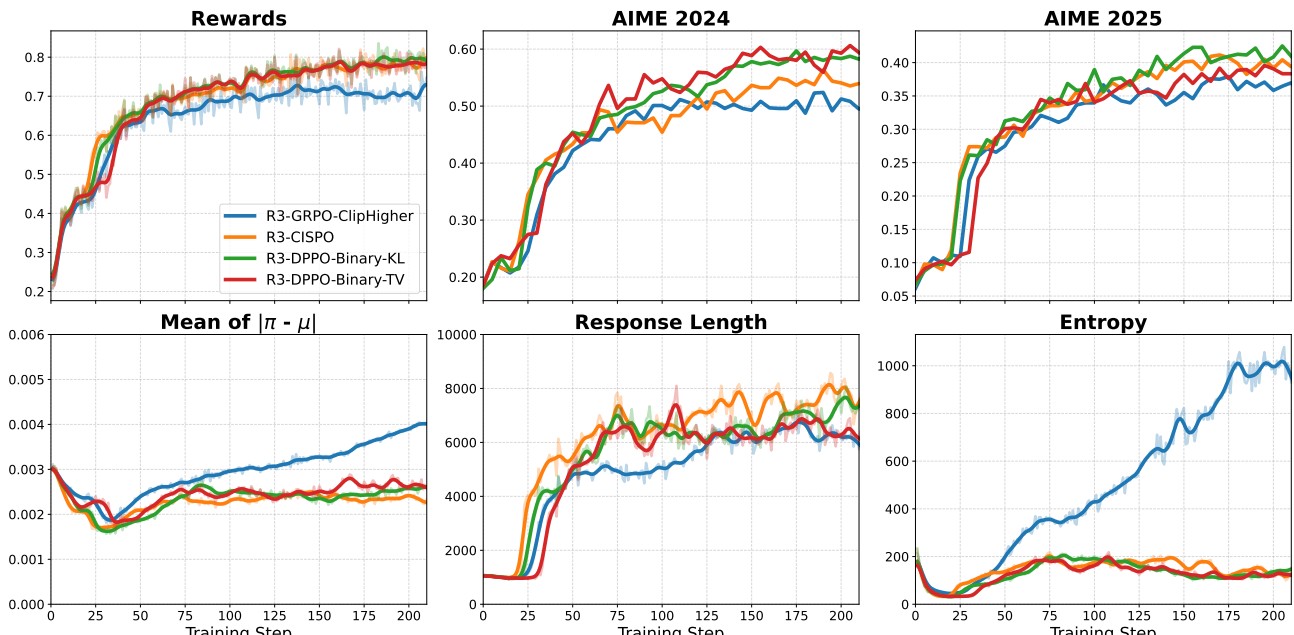

*Figure 13.* Evolution of metrics for **MoE Base w/ R3** experiment (based on Qwen3-30B-A3B-Base, with rollout router replay).

highlighting its *stability* and *efficiency*. On both AIME 24 and AIME 25 benchmarks, DPPO exhibits a clear, stable upward trend during training and maintains superior performance after convergence. The stability of our approach is evidenced by learning curves that generally show less fluctuation compared to baseline methods. Its efficiency is reflected in the rapid increase of training rewards and the strong final performance.

DPPO variants consistently demonstrate healthy training dynamics. The training-inference mismatch (measured by the mean absolute deviation $|\pi - \mu|$) and policy entropy remain within a stable, proper region throughout RL training. DPPO also effectively increases the generated response length across all scaling experiments, except for MoE Thinking. We note that the model Qwen3-30B-A3B already produces extremely long responses; as our training enforces a maximum length of approximately 16k tokens, RL training naturally shortens responses to fit this constraint.

In contrast, the GRPO-ClipHigher baseline, which relies on the ratio clipping mechanism of PPO, shows lower stability than DPPO and achieves inferior final performance in all five large-scale experiments. For example, in MoE Base w/o R3 (see Figure 12), GRPO-ClipHigher, though more stable than CISPO, improves more slowly and converges to lower training rewards and AIME scores than DPPO. In MoE Thinking (see Figure 14), GRPO-ClipHigher suffers a significant training collapse. Notably, GRPO-ClipHigher consistently leads to excessively high entropy in all large-scale experiments, a phenomenon not observed with other methods.

The CISPO baseline, which retains gradients for all tokens, is generally less stable and prone to collapse in certain settings. For instance, in MoE Base w/o R3 (see Figure 12), CISPO experiences a sudden and severe collapse leading to complete failure. In Dense Base (see Figure 15), CISPO shows a degenerative trend, particularly on AIME25. In MoE Base w/ LoRA (see Figure 16), the AIME24 scores, mean of $|\pi - \mu|$, and response length exhibit noticeable fluctuations, further indicating instability.

We also analyze the effect of rollout router replay (R3). Remarkably, DPPO variants *without* R3 already outperform baselines that use R3, underscoring the importance of a proper masking mechanism in RL training (see Figures 12 and 13). Furthermore, incorporating R3 yields additional gains for DPPO, suggesting that the benefits of R3 and DPPO are largely orthogonal. This implies that DPPO provides a robust foundation for LLM RL fine-tuning, capable of further improvement even when training-inference mismatch is mitigated by other techniques.

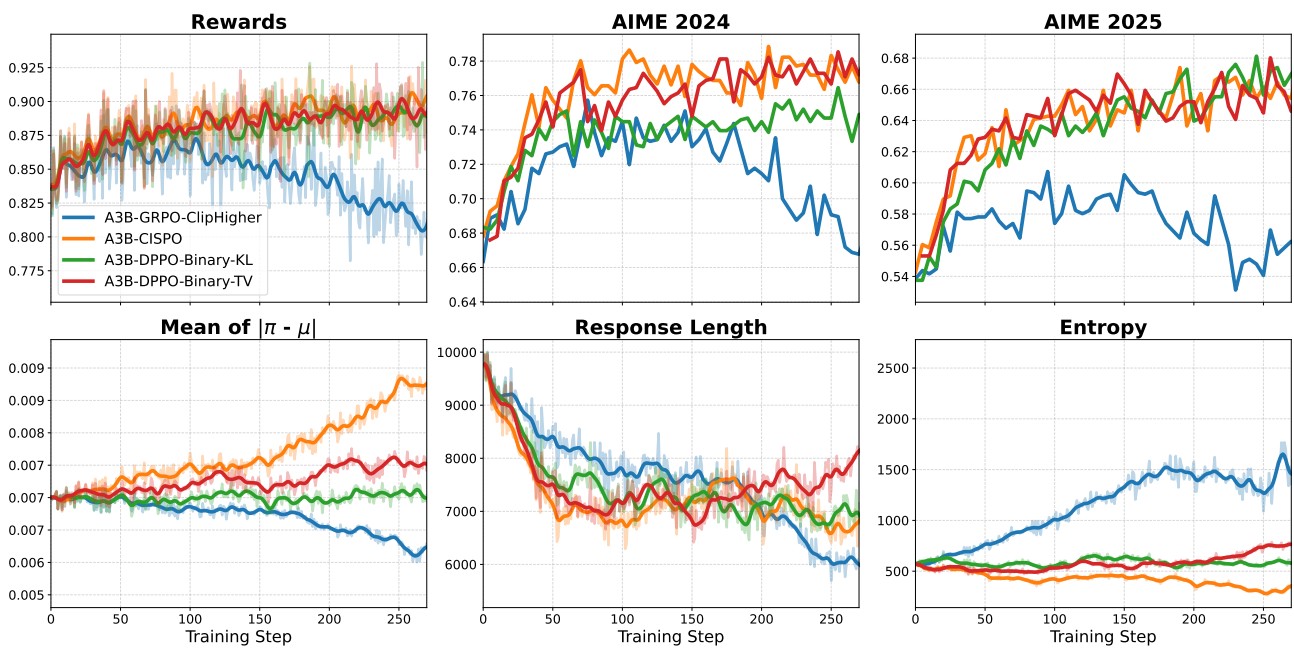

*Figure 14.* Evolution of metrics for **MoE Thinking** experiment (based on Qwen3-30B-A3B).

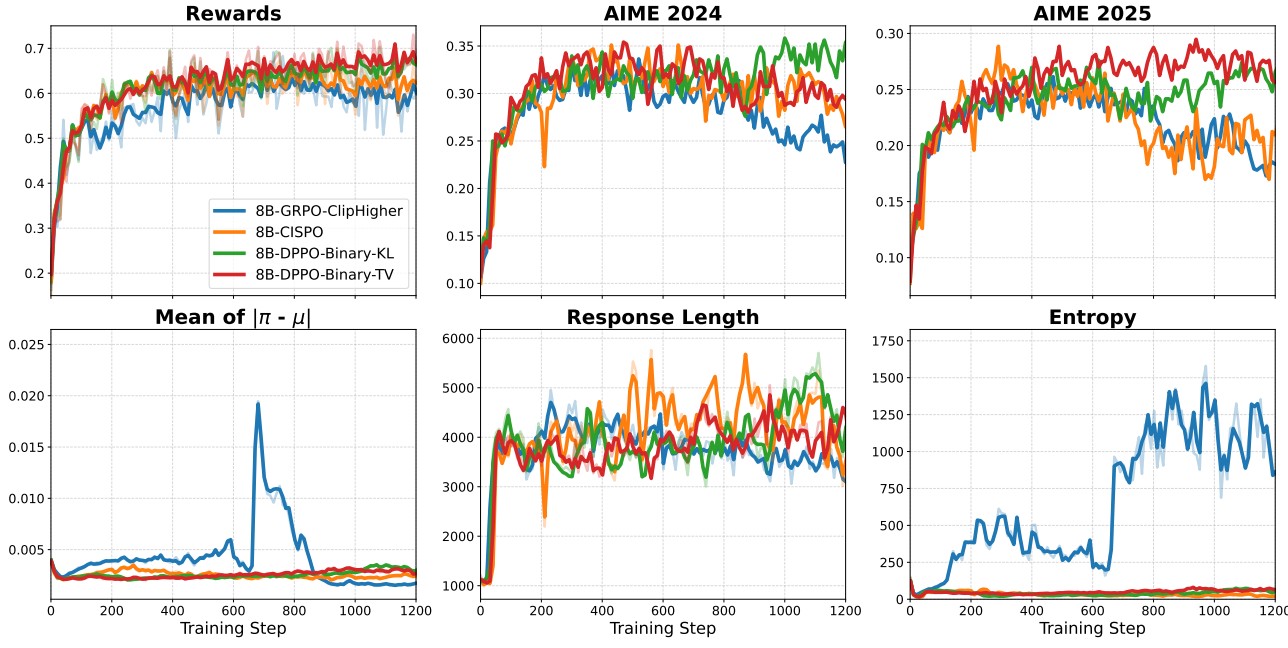

*Figure 15.* Evolution of metrics for **Dense Base** experiment (based on Qwen3-8B-Base).

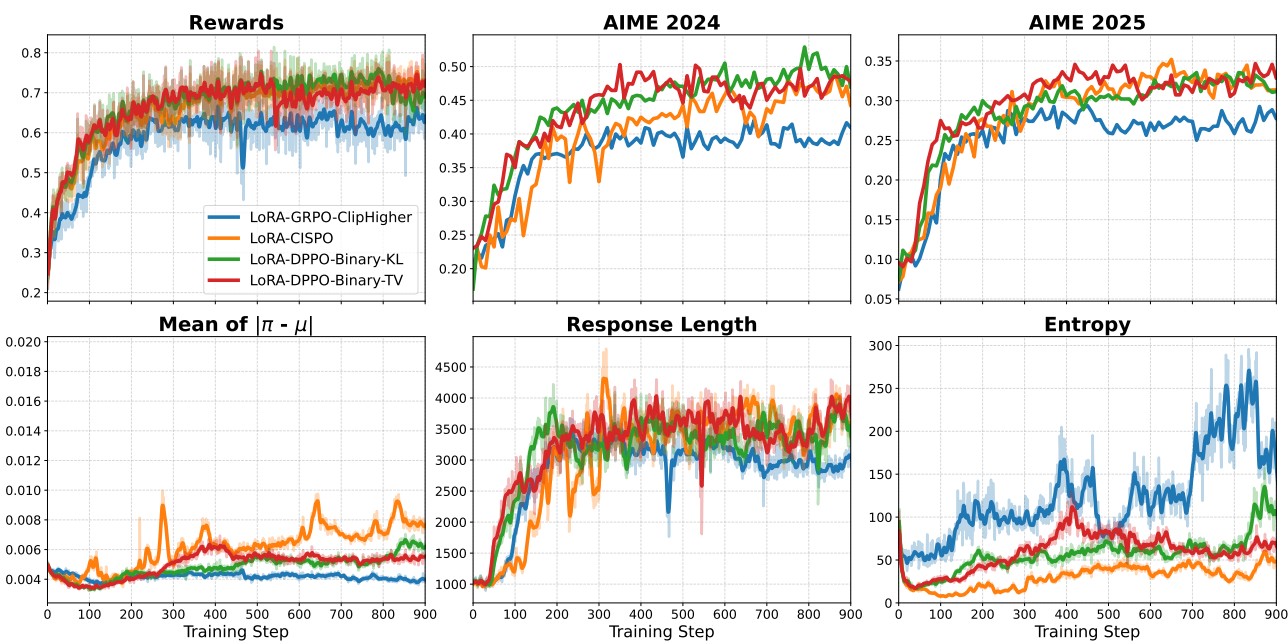

*Figure 16.* Evolution of metrics for **MoE Base w/ LoRA** experiment (based on Qwen3-30B-A3B-Base, with LoRA).

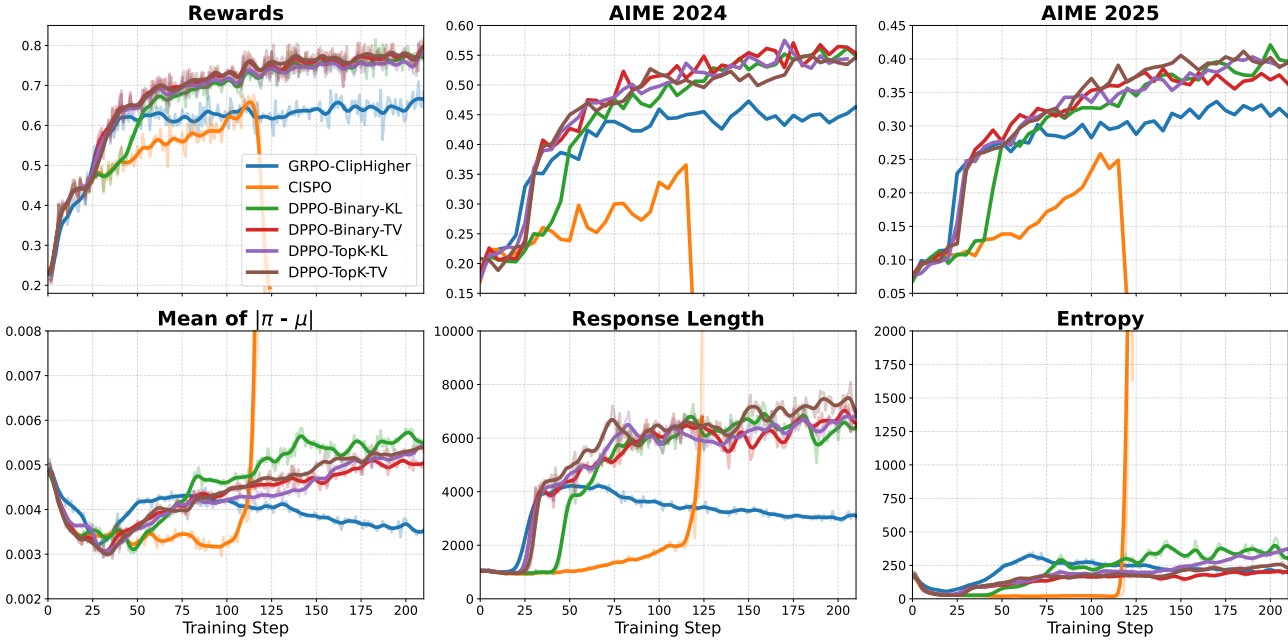

*Figure 17.* Evolution of metrics for baselines, DPPO with binary TV/KL approximation, and DPPO with Top-K (K=20) approximation under the same setting as MoE Base w/o R3.

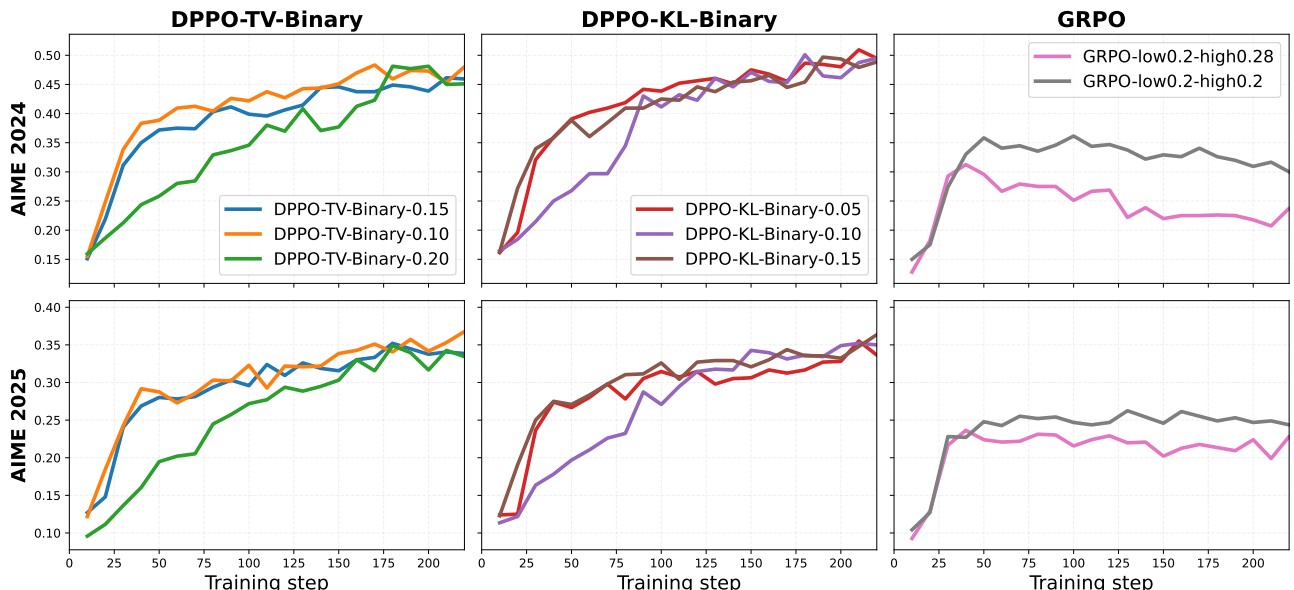

*Figure 18.* Hyperparameter sensitivity of DPPO-Binary-TV, DPPO-Binary-KL, and GRPO on Qwen3-30B-A3B-Base trained on DAPO with 8k context length. DPPO remains robust across a broad range of thresholds, while GRPO is more sensitive to the clipping range.

### G.3. Ablation on Divergence Approximation

In the scaling experiments, we compared DPPO variants using binary TV/KL approximations (Equations 13 and 14) against several baselines. To further investigate the approximation strategy, we experiment with DPPO variants with top-K TV/KL approximations (Equations 15 and 16), where we set $K = 20$; these variants are denoted as **DPPO-TopK-TV** and **DPPO-TopK-KL**. The choice $K = 20$ is limited by vLLM (Kwon et al., 2023), which supports returning log probabilities for at most 20 candidate tokens per step. We strictly replicate the experimental setting of MoE Base w/o R3. As in the main scaling experiments, for **DPPO-Binary-TV** and **DPPO-TopK-TV** we set the clip threshold $\delta = 0.2$, while for **DPPO-Binary-KL** and **DPPO-TopK-KL** we set $\delta = 0.05$.

As presented in Figure 17, introducing the top-K approximation does not yield significant performance gains, indicating that the simpler binary approximation already provides a sufficient and efficient proxy for constructing the trust region. This finding is encouraging, suggesting that DPPO with binary TV/KL remains highly scalable without sacrificing effectiveness.

### G.4. Hyperparameter Sensitivity

We further examine the sensitivity of the divergence threshold $\delta$ in DPPO. We fine-tune Qwen3-30B-A3B-Base on the DAPO dataset with an 8k context length and compare DPPO-Binary-TV with $\delta \in \{0.10, 0.15, 0.20\}$, DPPO-Binary-KL with $\delta \in \{0.05, 0.10, 0.15\}$, and GRPO variants using $\epsilon_{\text{low}} = 0.2$ with different upper clipping values.

As shown in Figure 18, DPPO is relatively insensitive within the tested ranges. DPPO-Binary-TV performs comparably for $\delta \in [0.10, 0.20]$, and DPPO-Binary-KL remains strong for $\delta \in [0.05, 0.15]$. In contrast, the GRPO curves vary more noticeably with the upper clipping parameter, and all tested DPPO configurations outperform the GRPO baselines.

### G.5. Extended Results for Different Model × Task Combinations

Besides experimental results presented in Section 7, we evaluate DPPO on more model × task settings to validate its advantage over the GRPO baseline. The settings we considered include:

1. **Different model family**. Training on a new model different from the Qwen family, OctoThinker-3B-Hybrid-Base (Wang et al., 2025b), on the standard math reasoning dataset (Hendrycks et al., 2021).

2. **Abstract reasoning and induction**. Training the Qwen3-1.7B-Base model on abstract reasoning task (Arc1D) and induction task (Acre) from the Gem library (Liu et al., 2025e).

3. **Multi-turn reasoning**. Training the Qwen3-1.7B-Base model on the multi-turn reasoning environment (Sudoku-v0-easy) from Gem the library (Liu et al., 2025e).

The training is conducted using Oat (Liu et al., 2025c) with their example scripts (thereby the standard hyper-parameters) for math RL and Gem RL. For the TV divergence clipping, we use a threshold of $\delta = 0.2$. Figure 19 shows the comparison between the TV variant of DPPO and the vanilla ratio-based PPO, both based on the GRPO algorithmic framework with the only difference being the trust region masking strategy. We can observe DPPO improves the efficiency (and sometimes asymptotic performance) over the baseline across different settings, validating its general effectiveness.

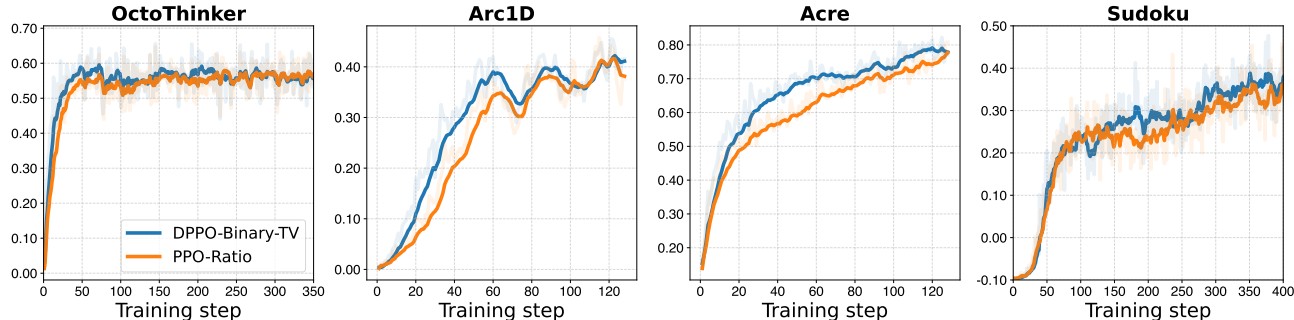

*Figure 19.* Learning curve comparison of using ratio (PPO-Ratio) and TV divergence (DPPO-Binary-TV) for the trust region clipping.

