# OpenReview forum: "Rethinking the Trust Region in LLM Reinforcement Learning"
_ICML.cc/2026/Conference — ICML 2026 regular_

### Official Review · Reviewer_3Hub · 2026-02-14

**Soundness:** 4
**Presentation:** 3
**Significance:** 4
**Originality:** 4
**Overall Recommendation:** 5
**Confidence:** 4

**Summary:**

This paper argues that PPO’s ratio clipping is structurally ill-suited for LLM reinforcement learning because the token-level ratio is a noisy single-sample proxy for true distributional change, which over-penalizes low-probability tokens and under-constrains high-probability mass shifts. To address this, the authors propose Divergence Proximal Policy Optimization (DPPO), replacing ratio-based clipping with a divergence-based trust-region mask grounded in a lower bound on a policy performance difference identity in the finite-horizon, undiscounted LLM setting.

**Compliance With Llm Reviewing Policy:**

Affirmed.

**Final Justification:**

This paper presents Divergence Proximal Policy Optimization (DPPO), an original, theoretically sound, and empirically validated solution to the structural limitations of PPO in LLMs. My initial review noted the rigorous methodology but raised clarity concerns regarding the role of off-policy data and the presentation of the theoretical bounds.

The authors’ rebuttal addressed these points. They strengthened their rationale for the necessity of off-policy optimization due to training-inference mismatches and committed to surfacing the linear bound in the main text. This response reinforced my initial assessment to confidently recommend acceptance.

**Key Questions For Authors:**

### Comments on Clarity

*Figure 1 clarity*: Please explicitly state that the TV distance shown is token-level D_TV (μ(⋅∣s_t)∥π(⋅∣s_t)), not trajectory-level, since that distinction matters for interpretation. While this can be inferred after reading the full paper, it is not at all clear in the introduction and may confuse the reader.

*Section 2.2 exposition*: A brief reminder that RL commonly uses off-policy data (buffer collected under μ) because data collection is expensive would make the derivation feel more natural. **In addition, a section on the role of off-policy data in LLM post-training is urgently needed.**

*Bound presentation*: Since Appendix B.3 derives a tighter, more practical linear/composite bound, consider surfacing it in the main text or motivating why the quadratic form is emphasized. Even more so since it seems that the linear bound more tightly matches the DPPO method since there divergences across all time-steps matter equally, not just the supremum.

*Objective bridge*: Eq. (9) could be connected more explicitly to the LLM surrogate L_μ^' (π) and the constrained form in Eq. (8), since the conceptual link is strong but currently somewhat implicit.

*Small notation issue*: In Eq. (10), ensure conditioning on s_t is consistent for μ on the RHS.

### Questions
1. What is the role of off-policy data in RL post-training. Alternatively, what is the role of trust-regions in on-policy optimization.
2. Concerning the binary and top-K approximations of DPPO, do I understand correctly that computationally nothing is gained compared to full DPPO (the logits anyhow need to be computed), but in case of off-policy data the memory consumption of caching the logits from the reference policy is dramatically reduced? Can I just run full DPPO in the on-policy case?
3. In the experimental setup you write: "In this setting, a stable algorithm
should theoretically converge to 100% training accuracy, as
all problems are known to be solvable by the initial model". Does this mean that the rollouts used for training are such that each problem is solved at least once?
4. In Takeaway 2 you write: "Using a recomputed
on-policy distribution as the anchor leads to instability. This
finding aligns with the theoretical bound in Equation (7)". How does this align with the bound of Eq. (7)? The bound works with any anchor, so why not recompute it based on a newer checkpoint?

**Limitations:**

yes

**Strengths And Weaknesses:**

**Principled theoretical foundation**: The paper derives a performance difference identity tailored to LLMs and a corresponding policy improvement lower bound, which directly motivates a trust-region constrained objective. The authors then identify PPO-style clipping as a single-sample approximation to this theoretically grounded objective.

**Clear diagnosis + solution**: The authors identify the PPO issue theoretically (ratio as a poor divergence proxy in long tails) and propose a theoretically consistent remedy (divergence-based masking). DPPO is not an ad-hoc tweak but rather directly reconciles theory and practice.

**Strong empirical confirmation**: Although the primary contributions of the paper are theoretical and methodological, the authors provide extensive empirical confirmation across a plethora of tasks and models. DPPO demonstrates notably improved stability and training efficiency at essentially zero overhead.

**Presentation**: The paper conveys its central messages rather well. However, a proper discussion on off-policy and on-policy learning is missing and would be particularly important, given that usually verifiable rewards in LLM post-training can be obtained rather inexpensively. Additional remarks on the presentation are provided below.

---

> ### Author Rebuttal · Authors · 2026-03-30
>
> We thank the reviewer for the positive feedback and suggestions for improvement. We respond to individual points from your review below.
>
> > `Figure 1 clarity` and `Small notation issue`
>
> Thank you for pointing them out, we will update the manuscript accordingly.
>
> > Q1: What is the role of off-policy data in RL post-training. Alternatively, what is the role of trust-regions in on-policy optimization.
>
>
> As detailed in Appendix A.2, RL post-training is inherently off-policy, even in nominally "on-policy" setups, due to training-inference mismatch. We highlight two main roles for off-policy data and trust regions in this context:
> - **Correcting System-Level Mismatches**: Modern RL frameworks use different engines for generation (e.g., vLLM/SGLang) and training (e.g., FSDP/Megatron). This implementation divergence introduces numerical disparities, causing the behavior distribution $\mu$ to diverge from the target distribution $\pi$, even with the same parameters ($\mu_\theta \neq \pi_\theta$)  [1]. Consequently, off-policy importance sampling is essential to correct the gradient. Trust-region methods are employed because they provide a much better bias-variance tradeoff than vanilla sequence-level importance sampling (which is unbiased but with high variance).
> - **Maximizing Training Throughput**: Autoregressive rollout generation is typically memory-bounded, thus expensive and slow. To improve efficiency, existing frameworks typically collect a large batch of data and split it into multiple mini-batches to perform multiple gradient updates [2]. This process inherently requires off-policy optimization, and trust regions are essential to ensure the policy does not collapse during these updates.
>
> [1] https://arxiv.org/pdf/2510.26788
>
> [2] https://arxiv.org/abs/2512.02556
>
>
> > Q2: Concerning the binary and top-K approximations of DPPO, ...? Can I just run full DPPO in the on-policy case?
>
> Yes, the computational gain is trivial, the main benefit is the reduction of memory overhead. But as explained above, due to the training-inference mismatch, LLM RL training is always in off-policy setting. The on-policy setting requires exactly the same distribution for behavior distribution ($\mu$) and target distribution ($\pi$), in this case, trust-region is not needed because the ratio $\frac{\pi}{\mu}$ is always 1.
>
> > Q4: How does this align with the bound of Eq. (7)? The bound works with any anchor, so why not recompute it based on a newer checkpoint?
>
> In Eq. (7), $\mu$ is the policy/distribution for data sampling. When we collect a batch of data by sampling from $\mu_{\theta'}$, a recomputed distribution $\pi_{\theta'}$ can be different due to the training-inference mismatch.
>
> > `Section 2.2 exposition`
>
> Thanks for the insightful suggestion. We will explicitly highlight this `off-policy` setting in the next version and make sufficient discussion in the main text.
>
> > `Bound presentation`
>
> Thank you for your scrupulous reading and thoughtful suggestion. We highly agree that the linear bound is tighter and more practical. Because of the page limitation, we have to put one into the appendix. The reason why we put the quadratic supremum version is, it highly resembles the original TRPO bound (by replacing $\frac{1}{1-\gamma}$ with $T$), which makes it easier to follow. As you suggested, we will consider to put both bounds into the main text for better presentation.
>
> > `Objective bridge`
>
> In Appendix B.4, we build a direct connection between the surrogates of Eq. (3) and Eq. (5), which can be also applied to Eq. (9). Basically, the surrogates in Eq. (5) and Eq. (9) have **the same gradients** if we ignore the `min` and `clip`. This is achieved by replacing $R(y)$ with $A(y)=R(y)-B(x)$ for variance reduction where $B(x)$ is the average reward given prompt $x$, and replacing $r_t - 1$ with $r_t$ (a known technique for first-order approximation [3]). Both operations do not change the expectation of gradients, thus Eq. (5) and Eq. (9) are equal from the view of gradient descent optimization.
>
> The `min` and `clip` are from PPO algorithm, which acts as an approximation of the trust region constraint in Eq. (8). This is a well known step (from TRPO Eq. (8) to PPO Eq. (9)), however, may be somewhat implicit. [3] shows that, Eq. (9) can be rewritten in the format of Eq. (11) and (12) where $D=r_t-1$ . **The core contribution of DPPO is to directly use $D_{\mathrm{TV}}$ or $D_{\mathrm{KL}}$ in Eq. (12), instead of noisy $r_t - 1$.**
>
> We will make this connection more clear in our later manuscript. Thank you for this feedback.
>
> [3] https://arxiv.org/pdf/2512.01374
>
> > Q3: Does this mean that the rollouts used for training are such that each problem is solved at least once?
>
> The dataset is collected before training. Specifically, we unroll 40 responses for each problem using the same model, and only keep problems where the initial accuracy is between 20% and 80%. In this way, all problems are known to be solvable by the initial model.

---

> > ### Author Rebuttal · Reviewer_3Hub · 2026-04-01
> >
> > I have read the author response and the authors have satisfactorily addressed my questions and all concerns regarding the presentation. I am reaffirmed in my decision to recommend acceptance.

---

> > > ### Author Response · Authors · 2026-04-02
> > >
> > > Dear Reviewer 3Hub,
> > >
> > > Thank you for your thorough review and valuable feedback, particularly the suggestions for improving the presentation of our work. We truly appreciate your positive assessment, which is very encouraging to our team.
> > >
> > > Best regards,
> > >
> > > The Authors

---

### Official Review · Reviewer_E3UA · 2026-03-12

**Soundness:** 4
**Presentation:** 4
**Significance:** 4
**Originality:** 3
**Overall Recommendation:** 6
**Confidence:** 4

**Summary:**

The paper argues a structural weakness in PPO for LLM fine-tuning: ratio clipping produces noisy per-token divergence estimates that over-penalize low-probability tokens while missing large shifts among high-probability ones. The proposed DPPO replaces ratio clipping with a mask derived from token-level TV or KL divergence, using Binary and Top-K approximations as tractable lower bounds, and derives a Policy Improvement Bound for finite-horizon autoregressive generation. Experiments on Qwen3-30B-A3B, dense models, and Llama on AIME24/AIME25 show DPPO improves stability and sample efficiency over GRPO-ClipHigher, CISPO, PG-IS, and MiniRL.

**Compliance With Llm Reviewing Policy:**

Affirmed.

**Final Justification:**

I appreciate the authors’ active and thoughtful response. Their clarifications and additional explanations have strengthened my understanding of the work. Overall, I believe this paper reaches the quality bar for an oral presentation.

**Key Questions For Authors:**

I point the concerns outlined in the Weaknesses section. Addressing those points would help elevate what is already a strong paper to an even more compelling contribution. I am happy to revisit my assessment based on the authors' responses and the perspectives shared by fellow reviewers during the discussion phase.

**Limitations:**

Yes.

**Strengths And Weaknesses:**

**Strengths:**

1. The policy improvement bounds are derived specifically for finite-horizon, undiscounted autoregressive generation (the main theorem and its corollary), rather than importing standard discounted MDP trust region results. This correctly accounts for the sequential token-generation structure, and the connection between token-level TV divergence and the DPPO mask follows naturally from this theory.

2. The observation that PPO's ratio clipping is structurally misaligned with large LLM vocabularies is well-argued and novel in this explicit form. The analysis of ratio behavior for low-probability versus high-probability tokens (the ratio analysis figure) gives clear evidence of the asymmetry. Unlike Clip-Higher and CISPO, which patch symptoms, this work targets the root cause.

3. DPPO shows consistent gains across five large-scale settings, and the Binary-TV/Binary-KL variants require only a few lines of code change. Integration into the high practical relevance of stable RL training for LLM reasoning further strengthen the contribution.

4. The paper has theoretical groundwork (the theoretical foundation section), diagnosis of PPO's deficiency (the PPO diagnosis section), stability analysis with practical guidelines (the stability analysis section), and large-scale experiments. The figures add real value -- particularly the ratio analysis figure (ratio vs. TV divergence volatility), the training-inference mismatch figure, and the instability source isolation figure.

5. The stability analysis in the stability section yields three concrete findings from controlled ablations: (i) a trust region remains necessary even at small learning rates, (ii) it must be anchored to the rollout policy rather than a recomputed one, and (iii) instability is driven primarily by a small subset of updates on negative-reward samples. The stability analysis figures support these conclusions convincingly.

6. The stabilization guidelines in the stability section have standalone value: maintain a trust region regardless of learning rate, anchor it to the rollout policy, and exercise caution with token-level importance sampling (TIS) since it can suppress signals from rare tokens.

**Weaknesses:**

1. The Binary and Top-K divergence estimates are justified as lower bounds, but the paper lacks analysis of how tight these bounds are across training. The argument that Top-K concentrates most probability mass is empirical, not formal; a bound on the approximation gap or empirical tracking over training would be valuable. As training sharpens the policy distribution, the approximation quality could shift systematically.

2. The divergence threshold delta is a critical hyperparameter, yet no systematic sensitivity analysis is provided. The paper acknowledges dynamic thresholding as future work, but practitioners replacing PPO's well-understood epsilon with delta need concrete guidance on tuning it across model sizes, vocabulary sizes, and task types. Without this, the practical advantage over simple ratio clipping is uncertain.

---

> ### Author Rebuttal · Authors · 2026-03-30
>
> We thank the reviewer for the positive feedback and suggestions for improvement. We respond to individual points from your review below.
>
> > The Binary and Top-K divergence estimates are justified as lower bounds, but the paper lacks analysis of how tight these bounds are across training. The argument that Top-K concentrates most probability mass is empirical, not formal; a bound on the approximation gap or empirical tracking over training would be valuable. As training sharpens the policy distribution, the approximation quality could shift systematically.
>
> As proved in Appendix C.1 (line 870), for Top-K TV divergence, the approximation gap can be upper-bounded by the total probability mass of the tail ('other' in Section 4.4), which is typically very small. To empirically validate this, we track the top-20 probability mass during the training of a RLHF task. It has an initial value of 99.4%, and then increases to 99.9% after 100 training steps.
>
> Please find the training curves in https://imgur.com/a/9haM5EO .
>
> > The divergence threshold delta is a critical hyperparameter, yet no systematic sensitivity analysis is provided. The paper acknowledges dynamic thresholding as future work, but practitioners replacing PPO's well-understood epsilon with delta need concrete guidance on tuning it across model sizes, vocabulary sizes, and task types. Without this, the practical advantage over simple ratio clipping is uncertain.
>
> Thanks for the valuable suggestion. We add additional experiments to present the hyperparameter sensitivity of DPPO-TV-Binary, DPPO-KL-Binary and GRPO, by fine-tuning Qwen3-30B-A3B-Base on DAPO dataset with 8k context length. In general, the performance of DPPO remains comparable within a wide range of $\delta$, for example, 0.1 ~ 0.2 for DPPO-TV-Binary and 0.05 ~ 0.15 for DPPO-KL-Binary. Conversely, GRPO is much more sensitive to the $\epsilon_{\text{high}}$. Notably, all of DPPO experiments clearly outperform GRPO.
>
> Find more details in https://imgur.com/a/Paijygu . Due to the limited time and compute budget during rebuttal, we early stop some experiments for quick evaluation. We will run complete experiments when updating the paper.
>
> > Addressing those points would help elevate what is already a strong paper to an even more compelling contribution. I am happy to revisit my assessment based on the authors' responses and the perspectives shared by fellow reviewers during the discussion phase.
>
> We truly appreciate your encouraging feedback. We hope our new experiments can resolve your concerns.

---

> > ### Author Rebuttal · Reviewer_E3UA · 2026-04-01
> >
> > After reading the author rebuttal, I find that the authors have satisfactorily addressed my questions and concerns through both their clarifications and the additional experiments. These revisions, particularly on presentation and clarity, have strengthened my confidence in the paper, and I will raise my score accordingly.

---

> > > ### Author Response · Authors · 2026-04-02
> > >
> > > Dear Reviewer E3UA,
> > >
> > >
> > > Thank you for your valuable and insightful feedback, which has greatly helped improve our work. We are pleased to have addressed your concerns, and sincerely appreciate the time and effort you dedicated to the review process.
> > >
> > > We also would like to thank you for raising the score. Your support means a lot to us!
> > >
> > > Best regards,
> > >
> > > The Authors

---

### Official Review · Reviewer_D6yb · 2026-03-12

**Soundness:** 2
**Presentation:** 2
**Significance:** 3
**Originality:** 3
**Overall Recommendation:** 4
**Confidence:** 3

**Summary:**

This paper argues that PPO/GRPO-style ratio clipping is a poor proxy for the trust region in LLM RL, especially under large long-tailed vocabularies, because token-wise ratios can over-constrain low-probability tokens while under-constraining high-probability mass shifts. To address this, the paper proposes DPPO, which replaces ratio-based clipping with divergence-aware masking using TV/KL divergence and lightweight binary/top-K approximations.

**Compliance With Llm Reviewing Policy:**

Affirmed.

**Final Justification:**

The paper presents a novel and practically useful idea, with sufficient empirical support despite some remaining limitations in claim strength. The rebuttal addressed my main concerns and improved my evaluation, so I am raising my recommendation to 4.

**Key Questions For Authors:**

See Weaknesses.

**Limitations:**

No. The paper should more explicitly acknowledge the theory-method gap, the heuristic nature of binary divergence, the limited baseline coverage, and the fact that the strongest conclusions are broader than what the current evidence fully supports.

**Strengths And Weaknesses:**

Strengths

1. The paper identifies a real and important weakness of ratio clipping in large-vocabulary LLM RL. In particular, the paper clearly explains why low-probability tokens can be over-constrained while large changes on high-probability tokens may remain under-penalized under standard PPO-style clipping.

2. DPPO elegantly bridges theory and practice by introducing lightweight Binary and Top-K approximations to bypass the memory bottleneck of computing exact policy divergence. It also demonstrates strong scalability on large-scale architectures up to 30B MoE.

Weaknesses

1. The paper’s theoretical framing is built around a trust-region-style objective defined over distributional divergence, but the actual method is not a principled optimizer of that objective. In practice, DPPO is implemented as a token-level masking heuristic that blocks certain updates under hand-designed conditions. As a result, the theory appears to motivate the method only at a high level, rather than directly justify the concrete algorithm being evaluated.

2. The method still feels heuristic, especially in its binary approximation, which is a very coarse proxy for the full distributional divergence. While this approximation is computationally attractive, it is not obvious that it preserves the relevant geometry of policy change well enough to support the paper’s stronger claims.

3. The authors make sweeping claims that divergence-based control should globally replace ratio clipping. However, empirical evidence is strictly limited to math and logical reasoning tasks (e.g., MATH, AIME) that use objective, deterministic reward signals. The paper lacks evaluations on standard open-ended alignment benchmarks (where reward models are notoriously noisy and subjective), making the baselines insufficient to support such generalized claims.

4. The mechanism claims are stronger than the evidence: the paper shows that masking certain updates helps, but does not fully prove these updates are the true primary cause of instability.

---

> ### Author Rebuttal · Authors · 2026-03-30
>
> We thank the reviewer for the valuable feedback. We respond to individual points from your review below.
>
> > W3: The paper lacks evaluations on standard open-ended alignment benchmarks (where reward models are notoriously noisy and subjective)
>
> Thanks for your valuable suggestion on the improvement. To address your concern, we conduct an additional RLHF experiment. Specifically, we use Skywork-Reward-Llama-3.1-8B as the reward model to fine-tune Qwen3-4B-Instruct-2507 model on UltraFeedback dataset and report the training rewards of DPPO-TV-Binary and GRPO. The results are really encrouaging, even beyond our expectation.
>
> We summarize the training rewards in below table. Please find the training curves in: https://imgur.com/a/9haM5EO
>
> |Step|DPPO|GRPO|
> |---|---|---|
> |0|-1.03|-1.03|
> |150|51.32|36.70|
> |450|70.27|45.24|
>
> We also evaluate the fine-tuned model (at step 150, we found the final checkpoints have lower scores due to reward hacking) in AlpacaEval 2.0. We present the results in below table. Notably, our DPPO achieves new **SOTA** in the [community leaderboard](https://tatsu-lab.github.io/alpaca_eval/).
>
> |Model|Length-controlled Win Rate|Win Rate|Average Length|
> |---|---|---|---|
> |Initial model|59.39|69.38|3147|
> |With GRPO fine-tuning|77.05|72.20|1756|
> |With DPPO fine-tuning|80.90|79.93|2003|
>
> Additionally, we tried different models (gemma-2-9b-it and Qwen3-4B-Instruct-2507) and datasets (UltraFeedback and hh-rlhf) to further validate this conclusion. Please find more training curves in: https://imgur.com/a/geW9wPU
>
> > W1: The paper’s theoretical framing is built around a trust-region-style objective defined over distributional divergence, but the actual method is not a principled optimizer of that objective. In practice, DPPO is implemented as a token-level masking heuristic that blocks certain updates under hand-designed conditions. As a result, the theory appears to motivate the method only at a high level, rather than directly justify the concrete algorithm being evaluated.
>
> As pointed by Reviewer 3Hub (Objective bridge), we believe this is a presentation gap (due to page limit), instead of a theory-method gap.
>
> In Appendix B.4, we build a direct connection between the surrogates of Eq. (3) and Eq. (5), which can be also applied to Eq. (9). Basically, the surrogates in Eq. (5) and Eq. (9) have **the same gradients** if we ignore the `min` and `clip`. This is achieved by replacing $R(y)$ with $A(y)=R(y)-B(x)$ for variance reduction where $B(x)$ is the average reward given prompt $x$, and replacing $r_t - 1$ with $r_t$ (a known technique for first-order approximation [1]). Both operations do not change the expectation of gradients, thus Eq. (5) and Eq. (9) are equal from the view of gradient descent optimization.
>
> The `min` and `clip` are from PPO algorithm, which acts as an approximation of the trust region constraint in Eq. (4) and (8). This is a well known step (from TRPO Eq. (8) to PPO Eq. (9)), however, may be somewhat implicit. [1] shows that, Eq. (9) can be rewritten in the format of Eq. (11) and (12) where $D=r_t-1$ . **The core contribution of DPPO is to directly use $D_{\mathrm{TV}}$ or $D_{\mathrm{KL}}$ in Eq. (12), instead of noisy $r_t - 1$.**
>
> In summary, DPPO is still built on the shoulders of PPO, but with a principled approximation. As acknowledged by reviewer 3Hub, **DPPO is not an ad-hoc tweak but rather directly reconciles theory and practice.**
>
> We will make this connection more clear in our later manuscript. Thank you for this feedback.
>
> [1] https://arxiv.org/pdf/2512.01374
>
> > W2: The method still feels heuristic, especially in its binary approximation, ..., it is not obvious that it preserves the relevant geometry of policy change well enough to support the paper’s stronger claims.
>
> As stated in the paper, Top-K is a theorectically sound approximation.
>
> As proved in Appendix C.1 (line 870), for Top-K TV divergence, the approximation gap can be upper-bounded by the total probability mass of the tail ('other' in Section 4.4), which is typically very small. To empirically validate this, we track the top-20 probability mass during the training of a RLHF task. It has an initial value of 99.4%, and then increases to 99.9% after 100 training steps. Please find the training curves in https://imgur.com/a/9haM5EO .
>
> As mentioned in main text (line 435) and Appendix G.2, we found that the binary variants are equally good empirically (see Figure 14), which provides a simple alternative in practice. Additionaly, it is easy to check the binary approximations can effectively resolve the limitations in Section 4.2.
>
> > the paper shows that masking certain updates helps, but does not fully prove these updates are the true primary cause of instability.
>
> As in Figure 4, masking certain updates make a collapse training become stable, we think this already empirically proves these updates are the primary (may not be the only) cause of instability.

---

> > ### Author Rebuttal · Reviewer_D6yb · 2026-04-03
> >
> > The rebuttal adequately addressed my main concerns, especially by clarifying the theory-to-method connection and providing additional empirical evidence in more realistic RLHF settings. Based on these clarifications and new results, I am raising my score to 4.

---

> > > ### Author Response · Authors · 2026-04-04
> > >
> > > Dear Reviewer D6yb,
> > >
> > > Thank you for your thoughtful feedback, which has been very helpful in improving our paper especially for the RLHF evaluation. We are pleased to have addressed your concerns and truly appreciate the time and effort you dedicated to the review process.
> > >
> > > We are also grateful for the score increase. Your positive assessment is greatly appreciated!
> > >
> > > Best regards,
> > >
> > > The Authors

---

### Decision · Program_Chairs · 2026-04-30

**Decision:**

Accept (regular)

**Comment:**

This paper revisits the trust region formulation for reinforcement learning in the context of LLM finetuning. The authors discuss whether PPO’s ratio clipping is an appropriate proxy for controlling policy updates in large, long tailed vocabulary settings. The central claim is that the commonly used token level ratio is a poor surrogate for true policy divergence, leading to systematically imbalanced updates that over-constrain low probability tokens while under-constraining large shifts in high-probability mass.

Reviewers agreed that the paper identifies a real and important issue. The analysis provides a clear and convincing diagnosis of PPO’s limitations in the LLM regime, supported by both theoretical reasoning and empirical evidence. In particular, the derivation of policy improvement bounds tailored to the finite horizon, undiscounted setting of autoregressive generation is a strong contribution, and the connection between divergence based trust regions and token level updates is well motivated. The proposed DPPO framework replaces ratio clipping with a divergence based masking mechanism and introduces efficient Binary and Top K approximations to make this practical at scale.

Empirically, the method demonstrates consistent improvements in both training stability and efficiency across a wide range of settings, including large scale models and RLHF style tasks. The results show that DPPO stabilizes training dynamics while also improving convergence speed and final performance compared to PPO style baselines. The analysis of instability sources, along with the resulting practical guidelines such as anchoring the trust region to the rollout policy and handling low-probability tokens, further strengthens the contribution.

At the same time, reviewers noted several limitations. The gap between the theoretical formulation and the implemented algorithm remains somewhat indirect, as DPPO relies on masking heuristics rather than directly optimizing the derived objective. The divergence approximations, particularly the binary variant, are coarse and their tightness is not fully characterized. In addition, the role of the divergence threshold hyperparameter is important in practice but not fully explored in the initial version. Some concerns were also raised about the scope of evaluation and clarity of presentation.

These concerns were largely addressed during the rebuttal. The authors provided additional experiments, including RLHF style evaluations, and clarified the connection between theory and method, as well as the behavior of the divergence approximations and hyperparameter sensitivity. These additions improved confidence in both the validity and practical applicability of the approach.

Overall, the paper presents a well motivated and practically impactful contribution. It offers a compelling rethinking of trust region design for LLM reinforcement learning, supported by solid theoretical insight and strong empirical evidence. Despite some limitations in approximation and presentation, the work advances understanding of RL optimization in LLMs and provides a useful direction for future research. My recommendation is to accept.